# Efficient Reasoning Models: A Survey

**Sicheng Feng**                     *sicheng@mail.nankai.edu.cn*
*National University of Singapore, Singapore*
*Nankai University, Tianjin, China*

**Gongfan Fang**                       *gongfan@u.nus.edu*
*National University of Singapore, Singapore*

**Xinyin Ma**                       *maxinyin@u.nus.edu*
*National University of Singapore, Singapore*

**Xinchao Wang**\*                     *xinchao@nus.edu.sg*
*National University of Singapore, Singapore*

**Reviewed on OpenReview:** *https://openreview.net/forum?id=sySqlxj8EB*

## Abstract

Reasoning models have demonstrated remarkable progress in solving complex and logic-intensive tasks by generating extended Chain-of-Thoughts (CoTs) prior to arriving at a final answer. Yet, the emergence of this "slow-thinking" paradigm, with numerous tokens generated in sequence, inevitably introduces substantial computational overhead. To this end, it highlights an urgent need for effective acceleration. This survey aims to provide a comprehensive overview of recent advances in efficient reasoning. It categorizes existing works into three key directions: (1) *shorter* – compressing lengthy CoTs into concise yet effective reasoning chains; (2) *smaller* – developing compact language models with strong reasoning capabilities through techniques such as knowledge distillation, other model compression techniques, and reinforcement learning; and (3) *faster* – designing efficient decoding strategies to accelerate inference of reasoning models. A curated collection of papers discussed in this survey is available in our GitHub repository: https://github.com/fscdc/Awesome-Efficient-Reasoning-Models.

## 1 Introduction

Recent reasoning-oriented models, or Large Reasoning Models (LRMs) (Guo et al., 2025; Jaech et al., 2024), have achieved remarkable performance on complex reasoning tasks by generating long Chain-of-Thoughts (CoTs), enabling effective problem-solving in domains such as mathematics and coding (Sprague et al., 2024). However, while LRMs significantly improve performance on reasoning tasks, they also cause substantial overhead. Compared to standard Large Language Models (LLMs), reasoning models lead to redundancy across multiple dimensions.

A salient characteristic of reasoning models is their tendency to overthink by generating excessively long reasoning chains (Chen et al., 2024c; Sui et al., 2025a), which has naturally motivated efforts to improve efficiency by shortening reasoning paths. Meanwhile, recent studies (Wu et al., 2025d; Yang et al., 2025c; Jin et al., 2024b) challenge the assumption that longer CoTs always lead to better performance, showing even negative returns. To address this kind of CoT length redundancy, a range of methods have been proposed: reinforcement learning (RL) with length penalty (Luo et al., 2025a; Aggarwal & Welleck, 2025), supervised fine-tuning (SFT) on variable-length CoT data (Ma et al., 2025; Xia et al., 2025), and prompt-driven strategies that either guide reasoning paths or route inputs to more efficient solutions (Ding et al., 2024;

---

\*Corresponding author

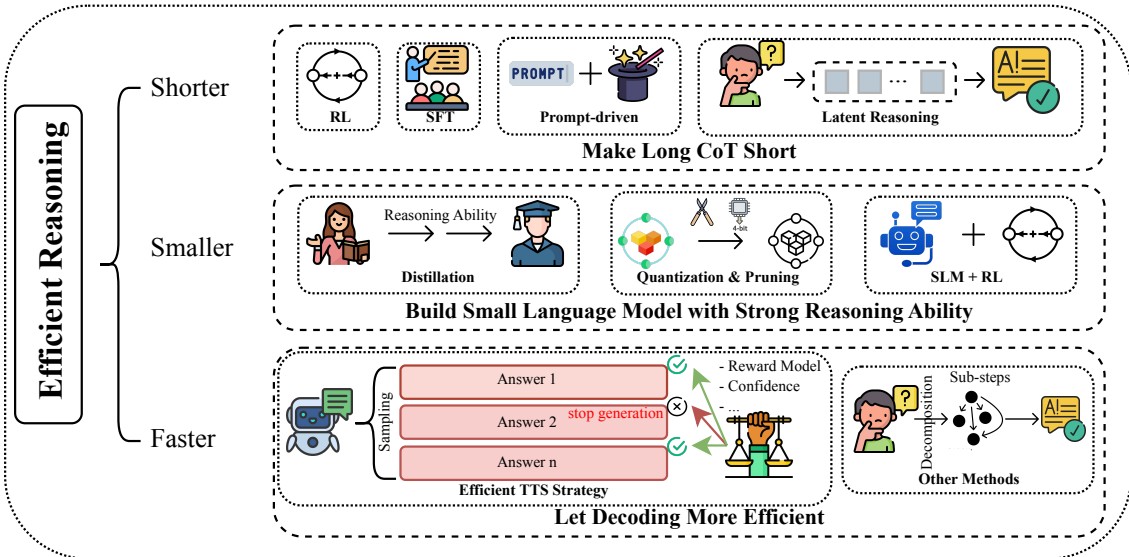

Figure 1: Overview of efficient reasoning. We categorize existing efficient reasoning methods into three key directions based on how they improve reasoning efficiency: (1) make long CoT short (*shorter*); (2) build small language models with strong reasoning ability (*smaller*); and (3) let decoding more efficient (*faster*).

Aytes et al., 2025). Furthermore, latent reasoning performs the process in latent space without generating explicit CoTs, making reasoning chains more concise (Hao et al., 2024; Su et al., 2025).

In addition to excessively long reasoning chains, reasoning models typically rely on large model sizes to achieve strong reasoning performance (e.g., DeepSeek R1 (Guo et al., 2025) has 685B parameters), which leads to substantial computational and memory costs. To address this, model compression (Han et al., 2016) has proven effective in reducing model size redundancy in standard LLMs, naturally inspiring interest in how these techniques (e.g., distillation (Hinton et al., 2015), quantization (Gray & Neuhoff, 1998), and pruning (LeCun et al., 1989)) can be applied to improve reasoning efficiency. In parallel, another line of work directly builds small language models with strong reasoning abilities using RL (Li et al., 2023a; 2025e; Zhu et al., 2024b).

Beyond length and model size redundancy, inefficiency can also arise during the decoding stage. A growing body of work focuses on accelerating inference through more efficient decoding strategies to tackle this issue. Test-time scaling (TTS) strategies, while enhancing reasoning performance (Snell et al., 2024), also introduce latency redundancy during the decoding stage. Some methods (Sun et al., 2024a; Wang et al., 2024b) specifically target and optimize the speed of certain TTS strategies (Wang et al., 2022a). Other approaches, like parallel decoding (Ning et al., 2023) and problem decomposition (Teng et al., 2025), also mitigate inefficiency.

This survey aims to provide an overview of research in efficient reasoning. As illustrated in Figure 1, we categorize existing works into three key directions based on the type of redundancy they target: (1) making long CoT short (*shorter*), which focuses on enabling models to produce shorter reasoning paths while maintaining performance; (2) building small language model with strong reasoning abilities (*smaller*), which aims to endow compact models with the ability to solve complex reasoning tasks; (3) making decoding more efficient (*faster*), which explores strategies to reduce latency during the decoding stage.

The following sections of this survey cover the content as outlined below. Section 2 will explore key backgrounds closely related to efficient reasoning. Section 3 will systematically introduce various methods and their relationships across three categories. Section 4 presents the evaluation metrics, as well as datasets and benchmarks. Section 5 will discuss the key challenges in the field and propose some potential future research directions, while Section 6 will conclude the survey. Additionally, Figure 2 illustrates the taxonomy of efficient reasoning methods discussed in this survey.

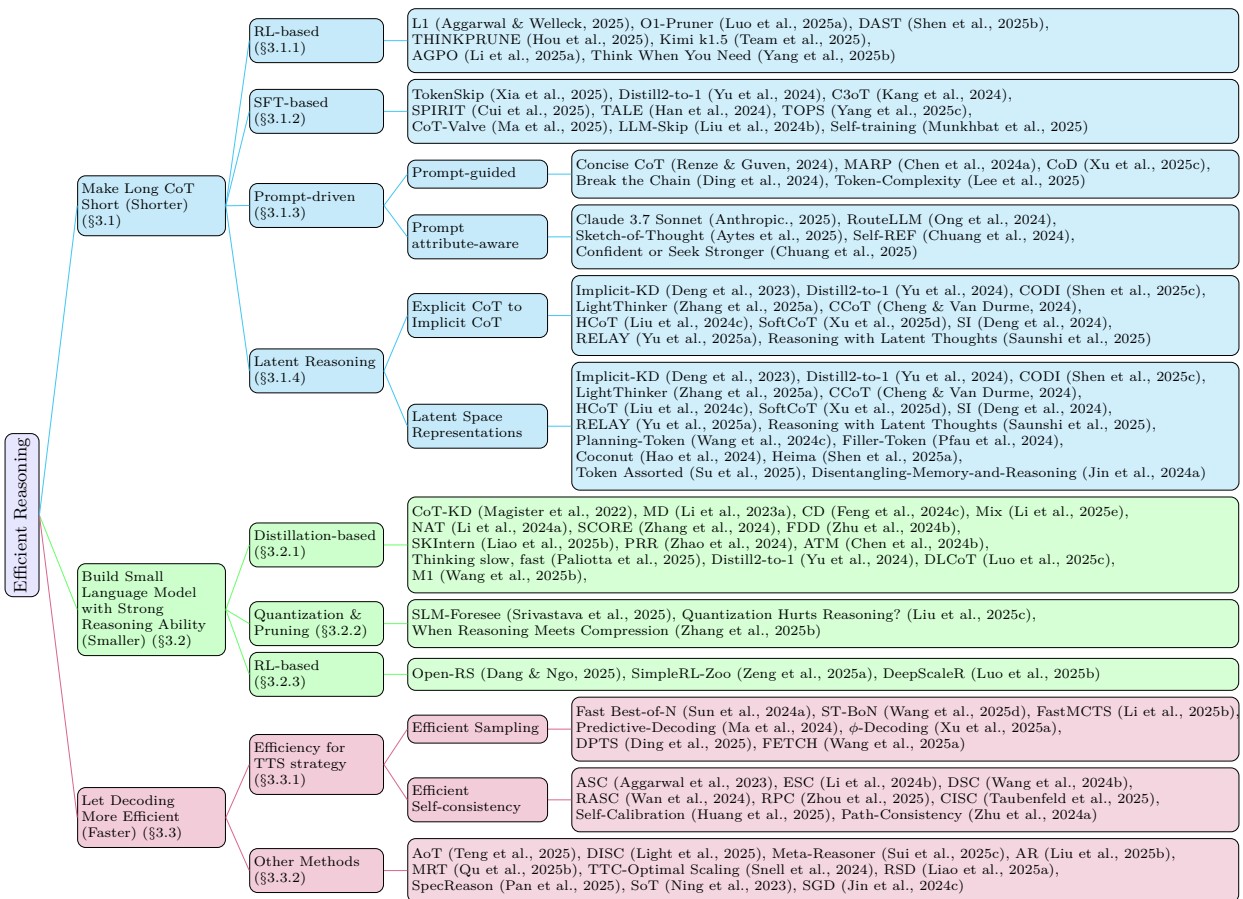

Figure 2: Taxonomy of efficient reasoning.

## 2 Background

### 2.1 Chain-of-Thought Reasoning

CoT (Wei et al., 2022) serves as a baseline reasoning approach, enabling LLMs to generate a sequence of intermediate steps before reaching the final answer, thus significantly improving performance on complex reasoning tasks. Various extensions have subsequently been proposed to further enhance reasoning capabilities. For instance, Tree-of-Thought (ToT) (Yao et al., 2023) generalizes the linear CoT structure into a tree, facilitating the exploration of multiple reasoning paths through backtracking and lookahead strategies. Graph-of-Thoughts (GoT) (Besta et al., 2024) has expanded this approach into graph structures to better capture dependencies and compositional relationships among reasoning steps, substantially improving reasoning quality. Additionally, some specialized CoT variants are task-specific. PoT (Chen et al., 2022) disentangles reasoning from computation by having the language model generate programmatic reasoning steps (i.e., expressing thoughts as code), which an external calculator executes to obtain the final answer, making this approach particularly effective for math and financial tasks. CoS (Hu et al., 2024), on the other hand, targets spatial reasoning by leveraging compressed symbolic representations of spatial relations to reduce token usage.

### 2.2 Reasoning Models and Underlying Techniques

Recent reasoning models have moved beyond early prompting-based CoT techniques by internalizing step-by-step reasoning through SFT and RL. Building structured reasoning paradigms mentioned in Section 2.1, these models are trained to generate reasoning traces aligned with human-like logic. RL plays a crucial

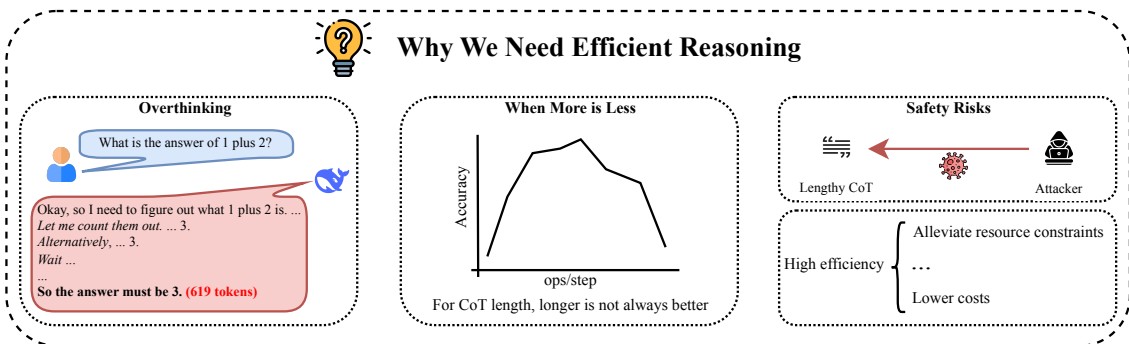

Figure 3: Motivation for efficient reasoning. *(Left)* Models often exhibit overthinking, generating unnecessarily long reasoning chains even for simple tasks. *(Middle)* Longer reasoning is not always better and may result in reduced accuracy when excessively verbose. *(Right)* Lengthy reasoning increases computational costs and poses safety risks. In addition, improving efficiency helps alleviate resource constraints and lower costs.

role by optimizing for reasoning quality using reward signals based on correctness, format alignment, and process supervision (Xu et al., 2025b; Ouyang et al., 2022; Zhou et al., 2023). Advanced models like OpenAI o1 (OpenAI, 2024) are believed to incorporate tree-search strategies (Coulom, 2006) and process reward models to guide the exploration of intermediate steps. Others, such as DeepSeek R1 (Guo et al., 2025), employ rule-based reward functions to reinforce correct reasoning steps.

## 2.3 Test-Time Scaling

Scaling test-time computation (TTC) is another road for enhancing reasoning performance (Snell et al., 2024; Zeng et al., 2025b). Scaling can be approached from two complementary dimensions: horizontal and vertical. The horizontal perspective involves generating multiple samples and selecting the best answer. Best-of-N (Cobbe et al., 2021; Sun et al., 2024a) selects the top-scoring response, while self-consistency (Wang et al., 2022a) identifies the most consistent answer across reasoning chains. The vertical perspective focuses on increasing the length of a single reasoning path. For example, Self-Refine (Madaan et al., 2023) iteratively improves an initial response via self-evaluation, while other works (Chen et al., 2024d; Gou et al., 2024) leverage external feedback to guide the refinement process. Additionally, an empirical study (Wu et al., 2025c) investigates the trade-offs between the efficiency and performance of various TTS strategies (e.g., Best-of-N, weighted voting) under different model sizes and computation budgets, providing practical insights for further research and deployment.

## 2.4 Model Compression

Model compression strategies are widely used to reduce the size and computational overhead of models (Han et al., 2016). Common approaches include quantization (Gray & Neuhoff, 1998; Frantar et al., 2023a; Lin et al., 2024; Xiao et al., 2023), which reduces model size by lowering the precision of model parameters. Pruning (LeCun et al., 1989; Ma et al., 2023; Fang et al., 2023; Wang et al., 2021) removes less significant or redundant model parameters to achieve sparsity, reducing model size and inference latency. Unlike the above techniques, knowledge distillation (Hinton et al., 2015; Wang et al., 2022b; Liu et al., 2019) achieves compression not by directly modifying the original model, but by transferring knowledge from a larger, well-trained teacher model to a smaller student model, allowing the student to replicate the teacher's behavior while maintaining comparable performance (see details about model compression in Appendix A.1).

## 2.5 Why We Need Efficient Reasoning

Efficiency is a valuable research direction across many fields, and in the context of reasoning, we highlight key motivations for pursuing efficient reasoning (see Figure 3). Reasoning models often generate excessively

Table 1: Performance of efficient reasoning methods on the AIME 24 dataset. † denotes the result of the original model, averaged over 5 independent runs.

| Category | Type | Methods | Acc. / #Tokens | Base Model |
|---|---|---|---|---|
| Original Model | - | Baseline† | 70.67% / 10024 | DeepSeek-R1-32B |
| Shorter | RL | DAST | 53.30% / 6337 | DeepSeek-R1-Distill-Qwen-7B |
| Shorter | SFT | CoT-Valve | 43.30% / 4630 | QwQ-32B-Preview |
| Shorter | SFT | TOPS | 46.00% / 6427 | Qwen2.5-32B |
| Smaller | KD | Mix | 10.00% / - | Qwen2.5-3B |
| Smaller | KD | DLCoT | 53.30% / 18825 | Qwen2.5-14B |
| Smaller | RL | Open-RS | 46.70% / - | DeepSeek-R1-Distill-Qwen-1.5B |
| Smaller | RL | DeepSacleR | 43.10% / - | DeepSeek-R1-Distill-Qwen-1.5B |
| Faster | Efficient self-consistency | RPC | 9.50% / - | InternLM-2-MATH-Plus 7B |
| Faster | Efficient sampling | $\phi$-Decoding | 16.67% / - | LLaMA3.1-8B-I |

long reasoning chains to solve reasoning tasks, even for simple samples, and typically rely on larger model sizes to achieve stronger reasoning performance. For example, answering "What is the answer of 1 plus 2?" requires 619 tokens from DeepSeek R1-685B (see Appendix A.2 for details). To further illustrate the overhead, we evaluated four versions of DeepSeek R1 on the AIME 24 dataset and observed consistently huge token counts: 15513 for 1.5B, 12377 for 7B, 10854 for 14B, and 10024 for 32B. Additionally, some strategies, such as Best-of-N and self-consistency, further scale the decoding process to enhance reasoning performance. These lead to substantial computational and memory demands. Moreover, overly long reasoning paths can accumulate errors and negatively impact final accuracy (Wu et al., 2025d; Yang et al., 2025c).

On the other hand, efficient reasoning is also essential in real-world applications such as embodied AI (Duan et al., 2022), agent systems (Wang et al., 2024a), and real-time platforms (e.g., autonomous driving (Cui et al., 2024)). In these scenarios, efficiency enables agents to process sensory inputs in real time, make swift and accurate decisions, and interact seamlessly with dynamic environments. Additionally, unnecessarily lengthy reasoning may increase safety risks (Kuo et al., 2025; Li et al., 2025d), posing unpredictable threats. These challenges collectively highlight the limitations of current reasoning models, underscoring the necessity of improving reasoning efficiency.

## 3 Efficient Reasoning

In the following, we introduce efficient reasoning methods based on three key categories: shortening long chains of thought, as discussed in Section 3.1; developing small language models with strong reasoning capabilities, details of which can be found in Section 3.2; and improving decoding efficiency, which is elaborated in Section 3.3. We present the performance of various efficient reasoning methods on the challenging AIME 24 dataset in Table 1 and further provide a latency-based summary of representative methods across categories on the GSM8K dataset in Table 5.

### 3.1 Make Long CoT Short

Recent works have explored various approaches to improve reasoning efficiency by shortening CoT length without compromising reasoning performance. Among them, RL with length penalty is widely used for encouraging concise and effective reasoning paths (see Section 3.1.1). Another line of work explores SFT with variable-length CoT data to improve reasoning efficiency, as discussed in Section 3.1.2. In addition, prompt-driven techniques improve reasoning efficiency by utilizing prompts, with further details available in Section 3.1.3. Finally, we explore latent reasoning, which performs the reasoning process in latent space and drastically reduces CoT length, with details provided in Section 3.1.4. Additionally, Table 2 provides an overview of these methods, showing that most RL-based methods utilize Full FT, while many SFT-based methods adopt Parameter-Efficient Fine-Tuning (PEFT) techniques like LoRA (Hu et al., 2022) to reduce cost. This trend suggests that RL-based methods require more extensive parameter updates, making lightweight adaptation less effective; for latent reasoning, Full FT remains dominant, and these methods

Table 2: Overview of efficient reasoning methods in Section 3.1. The speedup ratio is computed by comparing either the latency (L.) or the token count (T.). $Avg_1$ represents the average of Llama-3.2-3B, Gemma2-2B, Qwen2.5-3B, Qwen2.5-Math-1.5B, and DeepSeekMath-7B; $Avg_2$ represents the average of GPT-4o, GPT-4o-mini, Yi-lightning, o3-mini, and LLaMA3.1-8B-I.

| Type | Methods | Training Scheme | Acc. / #Tokens | Base Model | Speedup |
|------|---------|-----------------|----------------|------------|---------|
| RL | O1-Pruner | PPO (Freeze FT) | GSM8K: 96.50% / 543 | QwQ-32B | 1.5 - 2.0 × (L.) |
| RL | DAST | SimPO (Full FT) | MATH-500: 92.60% / 2802 | DeepSeek-R1-Distill-Qwen-7B | 1.6 - 2.2 × (T.) |
| RL | AGPO | GRPO (Full FT) | MATH-500: 77.20% / 463 | Qwen2.5-Math-7B | 1.3 - 1.5 × (T.) |
| RL | THINKPRUNE | GRPO (Full FT) | MATH-500: 83.90% / 2209 | DeepSeek-R1-Distill-Qwen-1.5B | 1.7 - 2.0 × (T.) |
| RL | Think When You Need | GRPO (Full FT) | - | - | 1.3 × (T.) |
| SFT | TokenSkip | SFT (LoRA) | GSM8K: 78.20% / 113 | LLaMA3.1-8B-I | 1.7 - 1.8 × (L.) |
| SFT | C3oT | SFT (Full FT) | GSM8K: 47.10% / - | LLaMA2-Chat-13B | 2.0 × (T.) |
| SFT | Self-Training | SFT (Full FT) | GSM8K: 78.07% / 176 | $Avg_1$ | 1.3 - 1.5 × (T.) |
| SFT | TALE | SFT / DPO (LoRA) | GSM8K: 78.57% / 140 | $Avg_2$ | 1.7 × (T.) |
| SFT | CoT-Valve | Progressive SFT (LoRA) | GSM8K: 95.40% / 289 | QwQ-32B | 2.6 × (T.) |
| Prompting | Concise CoT | Training-free | - | - | 1.9 - 2.0 × (T.) |
| Prompting | Break the Chain | Training-free | GSM8K: 74.22% / - | ChatGPT | - |
| Prompting | TALE-EP | Training-free | GSM8K: 84.46% / 77 | GPT-4o-mini | 4.1 × (T.) |
| Prompting | CoD | Training-free | GSM8K: 91.10% / 44 | GPT-4o | 4.7 × (T.) |
| Routing | RouteLLM | LLaMA3-8B Router | GSM8K: 74.82% / - | GPT-4 | 1.5 × (T.) |
| Routing | Sketch-of-Thought | DistillBERT Router | - | - | 3.6 × (T.) |
| Routing | Self-REF | SFT (LoRA) | GSM8K: 81.60% / - | LLaMA3-8B-I | 1.2 - 2.0 × (L.) |
| Latent reasoning | Implicit-KD | SFT (Full FT) | GSM8K: 20.00% / - | GPT-2 small | 8.2 × (L.) |
| Latent reasoning | SI | Progressive SFT (Full FT) | GSM8K: 30.00% / - | GPT-2 small | 4.0 - 11.0 × (L.) |
| Latent reasoning | CCoT | SFT (LoRA) | GSM8K: 17.90% / - | CCOT & DECODE | 10.4 - 24.5 × (L.) |
| Latent reasoning | SoftCoT | SFT (Freeze FT) | GSM8K: 85.81% / - | Qwen2.5-7B-I | 4.0 - 5.0 × (L.) |
| Latent reasoning | CODI | Self-distillation (LoRA) | GSM8K: 43.70% / - | GPT-2 small | 2.5 - 2.7 × (L.) |
| Latent reasoning | LightThinker | SFT (Full FT) | GSM8K: 90.14% / - | Qwen2.5-7B | up to 1.4 × (L.) |
| Latent reasoning | Coconut | Progressive SFT (Full FT) | GSM8K: 34.10% / 8 | GPT-2 | 3.0 × (T.) |
| Latent reasoning | Token Assorted | SFT (Full FT) | GSM8K: 84.10% / 194 | LLaMA3.1-8B | 1.2 × (T.) |

often yield higher speedups, indicating that implicit representations enable more effective compression and offer a higher upper bound compared to explicit reasoning chains.

### 3.1.1 Reinforcement Learning Helps Efficiency Improvement

Incorporating explicit chain length penalty into RL is a natural strategy for shortening reasoning chains (Team et al., 2025; Li et al., 2025a; Arora & Zanette, 2025). L1 (Aggarwal & Welleck, 2025) takes this further by introducing designated length-constraint instructions into the training data. O1-Pruner (Luo et al., 2025a) develops a specialized reward design by utilizing length and accuracy from a reference model as baselines, explicitly rewarding shorter reasoning paths and higher accuracy to ensure efficiency without sacrificing performance. DAST (Shen et al., 2025b) aims to achieve a balanced CoT (i.e., dynamically adjusting computational resources by allocating more reasoning steps to more challenging questions and fewer to simpler ones). Specifically, it proposes a Token Length Budget (TLB), defined as a weighted sum of the mean token count in accurate answers and a predefined upper bound on generation length to quantify problem difficulty, penalizing excessively verbose reasoning for simple questions while encouraging comprehensive reasoning for complex ones. THINKPRUNE (Hou et al., 2025) designs a length-aware reward function that only provides a reward if the correct answer is generated within a specified token budget. The model is trained using the Group Relative Policy Optimization (GRPO) algorithm with progressively tightened length constraints. Additionally, Think When You Need (Yang et al., 2025b) utilizes pairwise comparisons to generate rewards based on the relative length and accuracy of reasoning, guiding models to produce concise yet accurate solutions.

### 3.1.2 Supervised Fine-Tuning with Variable-Length CoT Data Helps Efficiency Improvement

Following a clear fine-tuning pipeline, we organize the discussion of this line of research into two stages: (1) how variable-length CoT data is constructed and (2) which SFT approach (i.e., standard or progressive) is adopted. For each work, we explicitly address these two questions to facilitate comparison and analysis.

**How variable-length CoT data is constructed?**   To construct variable-length CoT data, long reasoning chains are commonly generated by prompting LLMs with inputs, whereas the key challenge lies in obtaining the corresponding shorter reasoning chains. To address this, existing approaches generally fall into two categories. The first approach involves compressing existing long reasoning paths into shorter ones. For instance, TokenSkip (Xia et al., 2025) identifies and skips less important tokens based on their semantic contribution to the final answer. Distill2-to-1 (Yu et al., 2024) discards reasoning steps entirely, retaining only high-quality (input, answer) pairs through consistency-based filtering. C3oT (Kang et al., 2024) leverages GPT-4 as a compressor to shorten chain length by preserving essential reasoning details. Additionally, SPIRIT (Cui et al., 2025) uses perplexity to evaluate step importance, thus selectively compressing reasoning paths.

The alternative approach directly generates short reasoning paths. Self-training (Munkhbat et al., 2025) employs multiple sampling combined with few-shot prompting, selecting the shortest correct reasoning paths. TALE (Han et al., 2024) observes that LLMs naturally follow token budget constraints specified in prompts and introduces a binary search-based algorithm to identify the optimal token budget for generating concise reasoning paths. TOPS (Yang et al., 2025c) begins with a small set of o1-like responses (i.e., either generated by existing models or manually constructed) as seed data. Each response corresponds to a different level of reasoning effort. Using this data, it trains a tag model that learns to produce variable-length reasoning paths conditioned on effort-specific prompts, enabling the construction of diverse CoT data with controllable lengths. Inspired by model merging (Yang et al., 2024b), CoT-Valve (Ma et al., 2025) achieves chain length control by adjusting a specific direction of the parameter space, merging parameters from a base LLM with those of a reasoning-enhanced model of identical architecture[1]. Additionally, LLM-Skip (Liu et al., 2024b) manually shortens reasoning paths for complex datasets at the initial training stage, explicitly labeling prompts with "Solve it in n steps.". In the subsequent progressive SFT process, shorter reasoning paths generated by the model are continuously integrated into the training set.

**Which SFT approach is adopted?**   Most works adopt a standard SFT approach (Xia et al., 2025; Yu et al., 2024; Kang et al., 2024; Cui et al., 2025; Munkhbat et al., 2025; Han et al., 2024; Ma et al., 2025; Yang et al., 2025c), typically leveraging either LoRA (Xia et al., 2025; Ma et al., 2025) or full fine-tuning (Kang et al., 2024). Notably, C3oT (Kang et al., 2024) designs a conditioned training strategy, enabling the model to learn both long and short reasoning styles during training and generate concise reasoning paths at inference by simply appending a short condition in the prompt. TALE (Han et al., 2024) further explores DPO as an alternative fine-tuning objective, allowing direct control over the model's output preference.

Another line of work adopts progressive fine-tuning strategies (Liu et al., 2024b; Ma et al., 2025). LLM-Skip (Liu et al., 2024b) iteratively encourages the model to generate shorter reasoning paths and then merges the generated shorter paths into the training set for subsequent fine-tuning rounds, gradually reducing chain length. CoT-Valve (Ma et al., 2025) supports both standard SFT and two progressive strategies: CoT-Valve++ and CoT-Valve+P. CoT-Valve++ introduces a normalized path-length factor $\beta$, which is smaller for longer paths. During training, the model parameters are dynamically adjusted along a direction scaled by $\beta$, allowing the model to adapt to reasoning paths of varying lengths and learn finer-grained length control. CoT-Valve+P, on the other hand, progressively trains the model on samples sorted from long to short chains, guiding it to shorten the chain length over successive fine-tuning stages.

### 3.1.3   Prompt-Driven Efficiency Enhancement in Reasoning

We categorize prompt-driven works into two directions: (1) prompt-guided reasoning, which leverages well-designed prompts to guide reasoning models toward more effective reasoning paths and (2) prompt-based routing, which utilizes prompt-level attributes (e.g., complexity) to adaptively select appropriate computational paths (e.g., route easy questions to lightweight models and hard ones to powerful large models).

---

[1]Model merging is an effective strategy for efficient reasoning. For example, Kimi k1.5 (Team et al., 2025) improves token efficiency by merging a long-cot model and a short-cot model, while Wu et al. (2025a) combines System 1 and System 2 models to shorten response length.

**Prompt-guided Efficient Reasoning.** Concise CoT (Renze & Guven, 2024) shows that simply adding "Be concise" to the prompt can shorten reasoning chains. Break the Chain (Ding et al., 2024) leverages carefully crafted instructions (e.g., "rapidly evaluate and use the most effective reasoning shortcut") to trigger the model's ability to exploit shortcuts and skip unnecessary steps. TALE-EP (Han et al., 2024) employs an LLM-based estimator to predict the minimal token budget required for each question, which is then incorporated into the prompt to guide efficient reasoning. CoD (Xu et al., 2025c) develops the instruction "Think step by step, but only keep a minimum draft for each thinking step, with 5 words at most.", which significantly reduces token usage under few-shot settings without compromising accuracy. However, its performance degrades in zero-shot settings and on small language models. MARP (Chen et al., 2024a) boosts per-step information density and reduces step count under a fixed reasoning boundary, achieving high efficiency gains through prompt design, and can be further combined with PoT for better computation-reasoning separation. Token-Complexity (Lee et al., 2025) presents token complexity to measure the minimal tokens needed for correct reasoning and derives the theoretical compression limit of CoT chains. Through prompt variations (e.g., "use 10 words or less" or "remove all punctuation"), they explore the trade-off between performance and efficiency and show that current methods still fall far from the optimal bound, leaving room for improvement. Additionally, these methods can effectively construct variable-length CoT data, thereby supporting the approaches introduced in Section 3.1.2.

**Prompt Attribute-Aware Efficient Reasoning.** Claude 3.7 Sonnet (Anthropic., 2025) offers two response modes (e.g., quick answers or step-by-step thinking), allocating more compute to complex reasoning tasks. Although the implementation details remain undisclosed, it is the first hybrid reasoning model and a foundation for subsequent methods.

Routing strategies primarily fall into two categories: classifier-based and uncertainty-based. Classifier-based approaches train a separate router to categorize incoming questions and route them to the most suitable model. RouteLLM (Ong et al., 2024) trains a router using preference data to dispatch easy questions to lightweight and harder ones to stronger models. Sketch-of-Thought (Aytes et al., 2025) routes each input to the most appropriate reasoning pattern by referencing cognitive science (Goel, 1995), introducing three heuristic modes: Conceptual Chaining, which links ideas using minimal language; Chunked Symbolism, which organizes reasoning into symbolic blocks; and Expert Lexicons, which leverage domain-specific shorthand.

Uncertainty-based methods rely on confidence to guide routing. Self-REF (Chuang et al., 2024) adds two special tokens (i.e., <CN> for confident and <UN> for unconfident) to indicate confidence, training the model on annotated responses to self-assess its confidence level. If uncertain, the model defers to a more potent model or abstains. Confident or Seek Stronger (Chuang et al., 2025) further analyzes uncertainty-based routing, observing that uncertainty distributions are relatively stable across tasks but vary significantly across models and uncertainty quantification (UQ) methods. It further designs a calibrated data construction strategy that improves the reliability of routing decisions for small language models.

### 3.1.4 Reasoning in Latent Space

Unlike explicit CoT reasoning, latent reasoning (Deng et al., 2023; Tan et al., 2025) performs the reasoning process in latent space, skipping the generation of explicit intermediate steps. Latent reasoning brings two key benefits: it allows for more human-like thinking by modeling complex ideas beyond language, and improves efficiency by reducing the need for explicit reasoning chains. This section first examines how models transition from explicit to implicit reasoning. Then, we explore how reasoning is represented in latent space.

**From Explicit CoT to Implicit CoT.** As the seminal work introducing implicit CoT, Implicit-KD (Deng et al., 2023) proposed a distillation-based framework where a student model learns to reason implicitly by mimicking the hidden states across different layers of an explicit CoT teacher. To eliminate the reliance on the teacher model during inference, they further trained a simulator that directly maps input to teacher hidden states. SI (Deng et al., 2024) progressively removes intermediate reasoning steps through SFT, enabling the model to internalize reasoning without explicit chains. Similarly, Distill2-to-1 (Yu et al., 2024) showed that SFT on (input, answer) pairs alone can yield strong implicit reasoning capabilities. CODI (Shen et al., 2025c) introduces a novel self-distillation framework where a shared model acts both as teacher and

student—explicit CoT is learned via language modeling, while implicit CoT is learned by aligning the hidden activation of the token intermediately preceding the answer. LightThinker (Zhang et al., 2025a) proposes a dynamic compression strategy for CoT. It segments the reasoning chain and compresses each step into special tokens, with a focus on the KV cache compression. These latent representations are used for subsequent reasoning, with attention masks designed to ensure the model can only access compressed content rather than whole previous steps.

Another line of work explores using an auxiliary model to generate latent reasoning tokens directly from the input. CCoT (Cheng & Van Durme, 2024) trains a lightweight CCOT module (a LoRA (Hu et al., 2022)) to produce compressed latent reasoning tokens directly from input, which are then fed into a decoding module to generate concise answers, while HCoT (Liu et al., 2024c) adopts a similar pipeline but places greater emphasis on semantic alignment during compression. SoftCoT (Xu et al., 2025d) adopts a similar strategy by training a lightweight assistant model to produce implicit representations conditioned on the input. Furthermore, Reasoning with Latent Thoughts (Saunshi et al., 2025) demonstrated that looping a transformer multiple times could emulate a deeper model and naturally induce latent thoughts, effectively capturing iterative reasoning without tokenized steps. RELAY (Yu et al., 2025a) follows this idea by aligning each iteration of a looped transformer (Giannou et al., 2023) with explicit CoT steps. The trained looped model is then leveraged to produce high-quality CoT chains to train stronger autoregressive models on long reasoning tasks.

**Latent Space Representations for Reasoning.** A common choice for latent space representation is to use continuous tokens (Zhang et al., 2025a; Shen et al., 2025c; Cheng & Van Durme, 2024; Xu et al., 2025d; Hao et al., 2024; Liu et al., 2024c), which naturally align with the internal computation of neural networks. Coconut (Hao et al., 2024) models reasoning in the hidden space by feeding the final-layer hidden states back into the model without decoding explicit CoT tokens, enabling more continuous and efficient reasoning. This approach unlocks advantages that explicit CoT cannot offer, such as backtracking and parallel decoding. Inspired by Coconut, Heima (Shen et al., 2025a) introduces thinking tokens into multimodal large language models (MLLMs) to replace explicit reasoning steps, enabling reasoning in the latent space.

Another alternative approach is to employ discrete tokens as explicit representations of intermediate reasoning stages. Planning-Token (Wang et al., 2024c) employs a set of planning tokens inserted before each reasoning step to guide the model to generate a latent plan before producing the detailed explanation. These tokens are obtained by clustering the hidden states of reasoning steps, yielding semantically meaningful and distinct discrete representations. Filler-Token (Pfau et al., 2024) proposes inserting meaningless filler tokens (e.g., repeated dots) into the reasoning path, allowing the model to perform additional hidden computation, thereby enhancing performance on reasoning tasks. Token Assorted (Su et al., 2025) improves reasoning efficiency by mixing text tokens with latent tokens obtained through VQ-VAE (Van Den Oord et al., 2017), reducing sequence length while preserving key information. Disentangling-Memory-and-Reasoning (Jin et al., 2024a) introduces explicit discrete markers such as ⟨memory⟩ and ⟨reason⟩, which enable the model to disentangle reasoning into separate phases (i.e., retrieving relevant knowledge and performing logical inference) within the latent space. This separation facilitates more structured and interpretable reasoning behaviors.

## 3.2 Build Small Language Model with Strong Reasoning Ability

Compared to compressing reasoning chains, an alternative approach to improving reasoning efficiency is to empower small language models (SLMs) with strong reasoning capabilities. Due to their lower memory and computational requirements, SLMs are inherently more efficient and easier to deploy in real-world applications. Model compression (Han et al., 2016; Frantar et al., 2023b; Li et al., 2023b) naturally aligns with this goal, as it enables small or compressed models to retain or gain reasoning abilities. A natural starting point is to transfer reasoning capabilities from larger models via distillation (see Section 3.2.1). We further explore other model compression techniques, including pruning and quantization, which aim to compress models without severely compromising reasoning performance in Section 3.2.2. Beyond traditional model compression techniques, RL offers another promising direction, enhancing reasoning capabilities under limited resources through carefully designed training strategies, as discussed in Section 3.2.3. Additionally, a summary of these methods is presented in Table 3, indicating that most distillation approaches still rely

Table 3: Overview of efficient reasoning methods in Section 3.2. Blended$_1$ represents the combination of s1 and DeepSacleR datasets; Blended$_2$ represents the combination of Omni-MATH, AIME, AMC, and Still datasets.

| Type | Methods | Training Scheme | Training Data | Acc. | Base Model |
|------|---------|-----------------|---------------|------|------------|
| KD | CoT-KD | Distillation (Full FT) | CoT data | GSM8K: 21.99% (↑ 13.88%) | T5 XXL |
| KD | MD | Mixed distillation (Freeze FT) | CoT and PoT data | GSM8K: 41.50% (↑ 28.20%) | LLaMA2-7B |
| KD | Mix | Mixed distillation (Full FT & LoRA) | Long and short CoT data | GSM8K: 79.20% (↑ 1.70%) | LLaMA3.2-3B |
| KD | NAT | Mixed distillation (LoRA) | Positive and negative data | GSM8K: 41.24% (↑ 23.73%) | LLaMA-7B |
| KD | CD | Counterfactual distillation (Full FT) | Original and counterfactual data | - | - |
| KD | FDD | Feedback-driven distillation (Full FT) | Progressively add generated data | GSM8K: 49.43% (↑ 42.53%) | FlanT5-Large |
| KD | DLCoT | Distillation (Full FT) | High-quality data | GSM8K: 93.60% (↑ 9.10%) | LLaMA3.1-8B |
| KD | SKIntern | Distillation (LoRA) | Progressively simplify data | GSM8K: 33.90% (↑ 30.80%) | LLaMA2-7B |
| RL | Open-RS | GRPO (Full FT) | Blended$_1$ | AIME: 46.70% (↑ 17.80%) | DeepSeek-R1-Distill-Qwen-1.5B |
| RL | DeepSacleR | GRPO (Full FT) | Blended$_2$ | AIME: 43.10% (↑ 14.20%) | DeepSeek-R1-Distill-Qwen-1.5B |

on Full FT, with a few adopting PEFT techniques. Notably, methods that progressively incorporate refined or synthesized data (e.g., FDD and SKIntern) tend to achieve greater performance improvements.

Apart from model compression and RL, some studies explore the reasoning ability of small language models from alternative perspectives. For example, Liu et al. (2025d) shows that small language models can match or even surpass the reasoning performance of much larger LLMs with carefully designed TTS strategies. However, the effectiveness of TTS strategies varies with model architecture, reward design, and task complexity. While small language models show potential in reasoning, their limitations in instruction following and self-reflection highlight the need for further adaptation to align with human intent.

### 3.2.1 Distillation Transfers Reasoning Ability to Small Language Model

CoT-KD (Magister et al., 2022) first demonstrated that distillation can transfer reasoning ability from LLMs to small language models. However, due to limited capacity, small language models struggle to learn complex reasoning (Li et al., 2025e), motivating the development of more advanced strategies. Based on the optimization target, existing methods can be grouped into two directions: (1) data-focused, which improves the quality or composition of training data, and (2) model-focused, which concentrates on the distilled model itself or its generation strategy.

**Data-focused.** MD (Li et al., 2023a) adopts mix distillation by combining data generated with different prompting strategies (CoT and PoT) as training data, and Mix (Li et al., 2025e) applies a similar strategy using a mix of long and short CoT samples. CD (Feng et al., 2024c) enhances training diversity by mixing original data with counterfactual samples derived from it, while NAT (Li et al., 2024a) leverages negative data. DLCoT (Luo et al., 2025c) improves training data quality by segmenting and simplifying long reasoning paths. SCORE (Zhang et al., 2024) enables self-correction by allowing the model to generate, identify, and refine its reasoning, using the corrected outputs for further distillation. Distill2-to-1 (Yu et al., 2024) only retrains (input, answer) pairs as training data. The above methods rely on standard SFT, but some adopt progressive SFT. FDD (Zhu et al., 2024b) progressively adjusts data difficulty based on the small language model's performance on LLM-generated data, while SKIntern (Liao et al., 2025b) proposes a progressive process that removes symbolic knowledge and examples step by step, encouraging the model to internalize reasoning ability.

**Model-focused.** PRR (Zhao et al., 2024) distills two separate models: a probing model for retrieving relevant knowledge and a reasoning model for generating answers based on the question and retrieved content. Thinking slow, fast (Paliotta et al., 2025) explores distilling reasoning ability from transformer-based models into Mamba or Mamba-Transformer architectures to reduce inference cost. Similarly, M1 (Wang et al., 2025b) builds on Mamba (Gu & Dao, 2024) to develop a hybrid linear RNN reasoning model that alleviates latency and memory overhead from long reasoning chains, further enhanced through RL after distillation. Additionally, works such as NSA (Yuan et al., 2025) and MoBA (Lu et al., 2025), which focus on lightweight architectures for general efficiency, can also be extended to improve reasoning efficiency. Additionally, ATM (Chen et al., 2024b) designs an adaptive mechanism that enables the student model to

dynamically choose between pre-thinking (i.e., thinking before answering) and post-thinking (i.e., answering before thinking) based on question complexity.

### 3.2.2 Pruning or Quantization Retain Reasoning Ability

Recent work (Srivastava et al., 2025) systematically explores the impact of compression techniques like pruning and quantization on the reasoning capabilities of small language models, which shows that while quantization methods (Frantar et al., 2023b) have minimal impact on reasoning performance, pruning approaches (Li et al., 2023b) significantly degrade reasoning abilities. Similarly, When Reasoning Meets Compression (Zhang et al., 2025b) presents a comprehensive benchmark of compressed LRMs across various reasoning tasks. It also finds that quantized models retain strong reasoning performance and sometimes even surpass the original model, while aggressive pruning causes performance collapse at moderate sparsity. Furthermore, Quantization Hurts Reasoning? (Liu et al., 2025c) systematically evaluates the impact of quantization on reasoning models. It finds that high-bit (e.g., 8-bit) quantization is nearly lossless, while low-bit settings (e.g., 4-bit) significantly degrade performance, especially on complex tasks. Interestingly, the output length of CoT reasoning remains largely unchanged, except under aggressive quantization or when using small models. Notably, the results show that on certain large models, quantization can reduce GPU memory usage by over 75% while retaining nearly 100% of the original performance. Meanwhile, quantized versions of large models are often more effective than standalone small models, offering advantages in both memory efficiency and performance.

### 3.2.3 Reinforcement Learning Helps Build Small Language Model

SLM-Foresee (Srivastava et al., 2025) conducted a systematic study on the reasoning abilities of diverse small language models, demonstrating that small language models can exhibit strong reasoning potential. Certain models, such as the Qwen2.5 series (Yang et al., 2024a), even achieve performance comparable to or surpassing some LLMs. Open-RS (Dang & Ngo, 2025) enhanced the reasoning capability of small language models using RL with the GRPO algorithm (Guo et al., 2025) and curated a high-quality mathematical reasoning dataset derived from the s1 dataset (Muennighoff et al., 2025) and DeepScaleR dataset (Luo et al., 2025b). They further develop a cosine reward to control response length effectively. Their 1.5B model, trained on 7K samples within 24 hours on 4×A40 GPUs, achieved performance on benchmarks (e.g., AIME 24, MATH-500) that matches or surpasses models like o1-preview (AI, 2024). SimpleRL-Zoo (Zeng et al., 2025a) systematically evaluated the generality of ZeroRL (i.e., an RL paradigm that enables LMs to learn long-chain reasoning with only simple rule-based rewards and no additional supervision). The study proposed several key design strategies for successful ZeroRL training: using simple correctness-based rewards, aligning data difficulty with model capacity, and employing stable RL algorithms like GRPO. Remarkably, verification behavior was observed for the first time in small language models outside the Qwen2.5 series[2], further validating the reasoning potential of small language models. Additionally, DeepScaleR[3] (Luo et al., 2025b) leverages iterative scaling of GRPO to extend thinking length (i.e., 8K → 16K → 24K), significantly improving performance on math reasoning benchmarks. The 1.5B model, DeepScaleR-1.5B-Preview, surpasses o1-Preview and achieves 43.1% Pass@1 on AIME.

## 3.3 Let Decoding More Efficient

In the previous sections, we discussed two main directions for improving reasoning efficiency. However, this section covers strategies to accelerate reasoning during the decoding stage. It begins with techniques to reduce computational overhead during TTS (see Section 3.3.1), followed by an overview of other methods for making reasoning faster, with details provided in Section 3.3.2. These methods are summarized in Table 4, showing that most methods achieve notable efficiency gains and further improve model performance without additional training.

---

[2]Most existing works focus exclusively on Qwen2.5 models, whose strong instruction following and self-reflection abilities may skew results.

[3]DeepScaleR is a reasoning project for small language models, code and models are available at: `https://github.com/agentica-project/deepscaler`

Table 4: Overview of efficient reasoning methods in Section 3.3. The efficiency-up ratio is computed by comparing either the sampling count (S.), costs (C.), latency (L.), the correct trajectory count (T.), or FLOPs (F.). $C_1$ represents the consistency probability of the majority candidate. $C_2$ means the answer consistency within the sampling window. $C_3$ is the internal consistency via Chain-of-Embedding. $C_4$ is the probability of reaching the correct answer.

| Type | Methods | Training Scheme | Criteria | GSM8K $\Delta$ Acc. | Base Model | Efficiency-up Ratio |
|---|---|---|---|---|---|---|
| Efficient self-consistency | ASC | training-free | $C_1$ | 0.00% | GPT-3.5-Turbo | 1.4 - 4.3 $\times$ (S.) |
| Efficient self-consistency | ESC | training-free | $C_2$ | 0.00% | GPT-4 | 1.3 - 5.0 $\times$ (S.) |
| Efficient self-consistency | DSC | training-free | $C_1$ + Difficulty | $\downarrow$ 0.02% | GPT-4 | 2.6 - 5.0 $\times$ (C.) |
| Efficient self-consistency | Path-Consistency | training-free | - | $\uparrow$ 3.80% | LLaMA3-8B | 1.2 $\times$ (L.) |
| Efficient self-consistency | Self-Calibration | SFT (Full FT) | Confidence | $\uparrow$ 2.99% | LLaMA3.1-8B-I | 16.7 $\times$ (S.) |
| Efficient sampling | Fast Best-of-N | training-free | Reward score | - | - | 39.9 $\times$ (L.) |
| Efficient sampling | ST-BoN | training-free | $C_3$ | - | - | 2.0 $\times$ (L.) |
| Efficient sampling | FastMCTS | training-free | $C_4$ | $\uparrow$ 1.80% | Qwen2.5-7B | 1.1 - 3.0 $\times$ (T.) |
| Efficient sampling | Predictive-Decoding | training-free | - | $\uparrow$ 0.40% | LLaMA3-8B | - |
| Efficient sampling | $\phi$-Decoding | training-free | - | $\uparrow$ 6.14% | LLaMA3.1-8B-I | 2.8 $\times$ (F.) |
| Efficient sampling | Skeleton-of-Thought | training-free | - | - | - | 1.1 - 2.4 $\times$ (L.) |
| Other methods | AoT | training-free | - | $\uparrow$ 3.00% | GPT-4o-mini-0718 | - |

### 3.3.1 Efficiency for Test-Time Scaling Strategy

While TTS strategies (Snell et al., 2024) have shown great promise in improving reasoning performance without modifying model weights, they often cost significant computational overhead. To make TTS more efficient, we categorize this series of works into two directions: (1) efficient sampling methods that optimize the generation process in sampling-based TTS strategies and (2) efficient self-consistency techniques that reduce the cost of consistency-based reasoning.

**Efficient Sampling.** During the sampling process, the quality of generated reasoning chains often varies, and low-quality outputs lead to substantial redundant computation. A key challenge lies in how to allocate computation more effectively. A natural solution is to terminate low-quality outputs early. Fast Best-of-N (Sun et al., 2024a) proposes speculative rejection, which halts underperforming candidates based on early-stage partial rewards. ST-BoN (Wang et al., 2025d) adopts early consistency checks to identify and retain high-potential candidates while truncating the rest. Early path evaluation can also be applied to reasoning data synthesis. FastMCTS (Li et al., 2025b) leverages MCTS to build reasoning paths while evaluating quality at each step, allowing for dynamic path adjustment. Another line of work explores predicting the future trajectory to reduce redundancy and improve overall quality. Inspired by Model Predictive Control (Qin & Badgwell, 1997), Ma et al. (2024) proposes Predictive-Decoding, which mitigates the myopic nature of token-level generation in CoT by simulating several future reasoning steps (i.e., foresight trajectories) to reweight the token distribution. Similarly, Mendes & Ritter (2025) trains a value model from the language model's step-by-step generation dynamics to estimate the utility of intermediate reasoning states and decide whether to proceed. $\phi$-Decoding (Xu et al., 2025a) takes a step further by simulating multiple future paths at each step, clustering them to form a representative distribution and sampling the next step from this estimate.

Beyond token-level sampling, recent efforts have focused on structured sampling strategies within multi-path reasoning frameworks such as ToT and SoT. DPTS (Ding et al., 2025) proposes a Dynamic Parallel Tree Search framework that parallelizes reasoning path generation and dynamically manages cache states, enabling flexible path switching without deep exploration. It also incorporates early path evaluation to prioritize promising branches. Similarly, FETCH (Wang et al., 2025a) improves efficiency by merging semantically similar reasoning states to avoid redundant exploration and applying Temporal Difference (TD) learning (Sutton, 1988) with $\lambda$-return to stabilize verifier scores, reducing unnecessary switching.

**Efficient Self-Consistency.** Self-consistency also relies on repeated sampling, which leads to substantial computational overhead. Its core challenge aligns with efficient sampling—how to allocate computation adaptively. ASC (Aggarwal et al., 2023) estimates answer confidence during sampling and stops early once sufficient confidence is observed, while ESC (Li et al., 2024b) divides the sampling process into sequential

windows and stops sampling as soon as one window yields unanimous answers. DSC (Wang et al., 2024b) further incorporates difficulty awareness to better adjust the sample budget per instance. RASC (Wan et al., 2024) develops a similar early-stopping mechanism, terminating once sufficient high-quality samples are collected, followed by a score-weighted vote to determine the final answer. RPC (Zhou et al., 2025) combines self-consistency with perplexity-based estimation to accelerate convergence (i.e., the rate at which confidence estimation error for the final answer decreases with more samples). It also applies reasoning pruning to eliminate low-probability reasoning paths, reducing redundant computation. CISC (Tauben-feld et al., 2025) augments each sampled response with a model-predicted confidence score and performs confidence-weighted voting to improve final accuracy under the same sampling budget. Following the same idea, Self-Calibration (Huang et al., 2025) distills consistency signals from self-consistency into the model itself, enabling it to predict confidence scores during inference. This confidence is then used to guide early-stopping policies. Lastly, Path-Consistency (Zhu et al., 2024a) extracts high-confidence reasoning prefixes from early samples and reuses them to guide future sampling, improving generation speed and answer quality.

### 3.3.2 Other Methods for Making Reasoning Faster

One common approach is to decompose the original problem into sub-problems, reducing redundant token generation and skipping uninformative reasoning paths. AoT (Teng et al., 2025) constructs a DAG to model the dependencies among initially decomposed sub-problems. It then solves the overall task by iteratively decomposing and merging sub-problems. At each step, the model only processes a simplified version of the problem, reducing unnecessary token usage, minimizing attention overhead, and avoiding memory issues caused by long contexts. DISC (Light et al., 2025) dynamically partitions the problem into sub-steps and applies reward-based dynamic sampling and early stopping for each step to control compute costs, achieving efficient inference. AR (Liu et al., 2025b) decomposes the reasoning process into atomic reasoning actions organized into an atomic tree and performs structured reasoning via cognitive routing (e.g., reflection, backtracking, and termination). This atomic reasoning paradigm has also proven effective in multimodal large language models (MLLMs) (Xiang et al., 2025b). SoT (Ning et al., 2023) employs a two-stage decoding strategy by generating a reasoning skeleton and filling nodes in parallel. Inspired by SoT, SGD (Jin et al., 2024c) further builds a graph over sub-questions to capture logical dependencies and introduces difficulty-aware strategies to enable more efficient and higher-quality parallel decoding of reasoning models.

In real-world applications, LLMs are expected to adapt their output length to input complexity, producing detailed reasoning for complex tasks and concise responses for simpler ones. Several methods have been proposed to achieve this. TTC-Optimal Scaling (Snell et al., 2024) proposes a test-time compute-optimal scaling strategy that first estimates the difficulty of a prompt (i.e., either via oracle or model-predicted difficulty) and then adaptively selects different TTS strategies. For instance, on easy questions where the initial response is likely close to correct, self-verification is more efficient than multiple sampling; for complex problems, tree search with a verifier helps explore diverse reasoning paths. MRT (Qu et al., 2025b) further improves efficiency by introducing dense rewards based on reasoning progress (i.e., rewarding steps that increase the likelihood of reaching a correct answer) and training LLMs to progress toward solutions and avoid unnecessary computation. RSD (Liao et al., 2025a) enhances reasoning efficiency by combining a smaller draft model with a larger target model guided by a reward function. The draft model generates candidate steps, and if the reward is high, the output is accepted; otherwise, the target model refines it. Inspired by meta-cognition (Gao et al., 2024), Meta-Reasoner (Sui et al., 2025c) acts as a strategic advisor to guide the reasoning process, evaluate reasoning progress, and provide high-level guidance (e.g., backtracking, restarting) based on task complexity. Additionally, SpecReason (Pan et al., 2025) leverages the semantic tolerance in reasoning processes by using a lightweight model to speculate intermediate steps while reserving the large model for verification and correction.

### 3.4 A Supplementary: Intersections and Synergies Across Efficient Strategies.

Efficient reasoning strategies are not isolated, many methods combine ideas across categories to achieve better performance and flexibility. Distillation, beyond transferring reasoning capabilities, also serves as an effective means to realize latent reasoning (Deng et al., 2023; Shen et al., 2025c; Yu et al., 2024). Its core idea further supports SFT-based methods by enabling the student model to mimic multi-step reasoning

patterns (Kang et al., 2024; Munkhbat et al., 2025). Additionally, SFT and RL can be combined for adaptive reasoning. SFT is used to teach the model different answering modes, while RL helps the model learn when to switch among them based on input difficulty (Fang et al., 2025; Wu et al., 2025b).

## 4 Evaluation and Benchmark

### 4.1 Metrics

Assessing reasoning efficiency requires diverse metrics reflecting computational costs and model performance (e.g., accuracy). These metrics provide insights into the trade-offs between computational efficiency and model capability, moving beyond traditional evaluation methods that solely focus on performance by incorporating additional criteria such as token count, model size, and inference latency. In the following paragraphs, we present metrics for evaluating reasoning efficiency from both general and reasoning-specific perspectives. For the general perspective, we focus on metrics related to memory, computation, and power. For the reasoning-specific perspective, we first review classic metrics used to assess reasoning capability and then discuss metrics tailored specifically for reasoning efficiency.

#### 4.1.1 General Perspective

**Memory.**

- **Model Size** is a critical factor influencing its storage requirements and computational demands. It is commonly measured in megabytes (MB) or gigabytes (GB) and is particularly important for deployment in resource-constrained environments. Several key factors contribute to a model's size, including parameter count, data type, and specific architectural design choices.

- **Memory Footprint** refers to the amount of Random Access Memory (RAM) required to run a model during training or inference. This metric is essential for understanding the model's resource demands, particularly in environments with limited memory capacity, such as edge devices or lightweight servers. Memory is measured in units like MB or GB and is primarily determined by the model size and additional temporary data (e.g., intermediate variables).

**Computation.**

- **Floating Point Operations (FLOPs)** measures the number of floating-point arithmetic operations a model performs during inference or training. This metric reflects a model's computational complexity and is commonly used to assess its efficiency.

- **Latency (i.e., inference time)** measures the time required for an LLM to generate a response after receiving an input. This metric reflects the model's responsiveness and is particularly important in real-world applications (e.g., chatbots) where timely outputs are essential. Latency is typically measured in seconds (s) and depends on hardware capabilities, model size, and system optimizations. Additionally, latency can be evaluated in two key ways: end-to-end latency, which measures the total time from receiving an input to producing the final output, and next-token latency, which assesses the time required to generate each token in autoregressive models.

- **Throughput** measures an LLM's efficiency by the number of tokens generated per second, typically expressed as tokens per second (TPS). It indicates overall processing capability and is crucial for batch processing or large-scale deployments. For concurrent request scenarios, throughput can be expressed as queries per second (QPS).

**Power.**

- **Power Cost** refers to the total energy consumed by an LLM throughout its lifecycle, typically measured in Watt-hours (Wh) or Joules (J). It reflects the energy usage of key hardware components such as GPUs, CPUs, and DRAM.

- **Carbon Emission** measures the environmental impact of LLMs by quantifying the greenhouse gases produced during their life cycle. It is typically expressed in kilograms (kg) or tons of $CO_2$ equivalent ($CO_2$eq) and is influenced by factors such as hardware efficiency and model runtime. Carbon emissions can be estimated as follows (see Appendix A.4.1 for the formula). Several tools[4] are providing real-time emission tracking (e.g., CodeCarbon (Schmidt et al., 2021) and Carbon-Tracker (Anthony et al., 2020)) and predicting environmental costs (e.g., MLCO2 Impact (Lacoste et al., 2019)).

### 4.1.2 Reasoning-specific Persective

For reasoning evaluation, several accuracy variants are used. For example, greedy accuracy measures the accuracy when decoding deterministically (i.e., selecting the most likely token at each step). Minimum-maximum spread (Atil et al., 2024) quantifies stability by computing the accuracy gap across multiple runs. To better evaluate potential performance, the widely used Pass@k, which was initially proposed for generated code (Chen et al., 2021), has been adopted for reasoning tasks (Luo et al., 2023; Yu et al., 2023). It measures the probability of obtaining at least one correct answer among $k$ independent model outputs (see Appendix A.4.2 for the formula).

To capture stability, Pass∧k (Yao et al., 2024) is proposed, which measures the probability that all $k$ generations are correct (see Appendix A.4.3 for the formula). Pass∧k forms the basis for G-Pass@$k_\tau$ (Liu et al., 2024a), which further incorporates a tolerance threshold $\tau$, requiring only a minimum proportion of correct responses among the $k$ outputs. Furthermore, to jointly assess potential and stability, mG-Pass@$k_\tau$ interpolates G-Pass@$k_\tau$ over the interval $[0.5, 1.0]$, producing a comprehensive metric (see Appendix A.4.4 for formulas).

These metrics provide a complete view of LLM reasoning performance, balancing one-shot potential with consistency across trials. Additionally, Total Agreement Rate@N (TAR@N) (Atil et al., 2024) evaluates the consistency of a model by running it N times and measuring how often it produces identical outputs. It has two variants: TARa@N, which checks for agreement in the final answers, and TARr@N, a stricter version that requires an exact string-level match of the full outputs across runs.

To assess reasoning efficiency, token count (i.e., the number of output tokens generated by the model) is commonly used as an evaluation metric. Some studies have proposed composite metrics that integrate multiple dimensions of reasoning efficiency. CoT-Valve (Ma et al., 2025) proposes Accuracy per Computation Unit (ACU), calculated as accuracy divided by the product of parameter count and token count, explicitly considering the trade-offs among reasoning path length, model size, and model performance. Chen et al. (2024c) proposes two metrics: the outcome efficiency metric and the process efficiency metric (see Appendix A.4.5 for formulas). The outcome efficiency metric evaluates the proportion of efficient tokens (i.e., the tokens used until the first correct answer is produced) in the model-generated outputs. In contrast, the process efficiency metric assesses the diversity of reasoning paths within generated solutions.

Additionally, Cuadron et al. (2025) introduced the overthinking score, a reliable metric explicitly designed for quantifying the degree of overthinking in LLMs. The score is obtained using an LLM-based evaluator combined with structured prompt templates. Chen et al. (2024a) proposed the reasoning boundary (RB) to quantify the upper limit of LLM capability in handling complex reasoning tasks (see Appendix A.4.6 for the formula). Wang et al. (2025e) proposed the underthinking metric to evaluate whether a model prematurely abandons effective reasoning paths in incorrect responses, resulting in a large number of unproductive tokens (see Appendix A.4.7 for the formula).

**Preference for Metrics: Trade-off between Performance and Efficiency.** In most efficient reasoning studies, performance and efficiency are typically evaluated separately—performance is measured by accuracy or Pass@k, while efficiency is assessed via token count, latency, or model size. This decoupled evaluation is simple and effective. However, some recent works have proposed unified metrics that jointly capture both aspects. For example, CoT-Valve (Ma et al., 2025) introduces ACU, which combines parameter count, token count, and accuracy into a single metric. TALE (Han et al., 2024) proposes the optimal token budget, defined

---

[4]An online calculator: `https://mlco2.github.io/impact/`

as the minimum number of tokens required to maintain correctness, and uses search algorithms to guide the model toward more efficient reasoning. Moving forward, there is a growing need for better evaluation metrics that can balance performance and efficiency more holistically and practically. O1-Pruner (Luo et al., 2025a) proposes a novel metric called the Accuracy Efficiency Score (AES), which considers both the solution length and model accuracy and penalizes accuracy degradation more than it rewards improvement (see more details in Appendix A.4.8).

## 4.2 Datasets and Benchmarks

Datasets and benchmarks are crucial in evaluating language models' reasoning capabilities and efficiency. They provide standardized protocols for assessing how well models can perform reasoning tasks under various resource constraints, such as limited computing or inference budgets. These resources cover a broad spectrum of reasoning types—including mathematical, logical, and multi-hop reasoning—enabling comprehensive evaluation across diverse domains and difficulty levels (see more details in Table 6).

**Datasets.** To evaluate LLM reasoning ability, researchers commonly utilize developing reasoning benchmarks and datasets. Datasets are commonly categorized based on underlying reasoning types (Parashar et al., 2025), such as math reasoning (e.g., GSM8K (Cobbe et al., 2021), PRM800K (Lightman et al., 2023), MATH & MATH-500 (Hendrycks et al., 2021), AIME, and AQuA (Ling et al., 2017)), logical Reasoning (e.g., ProntoQA (Saparov & He, 2023)), common sense reasoning (e.g., StrategyQA (Geva et al., 2021), Hot-PotQA (Yang et al., 2018)), algorithmic reasoning (e.g., Game of 24 (Yao et al., 2023), Bin Packing (Parashar et al., 2025)), and planning (e.g., BlocksWorld (Valmeekam et al., 2023), Rubik's Cube (Ding et al., 2023), Trip Plan, and Calendar Plan (Zheng et al., 2024)).

**Benchmarks.** Sys2Bench (Parashar et al., 2025) is a benchmark suite designed for evaluating LLMs, comprising 11 datasets that cover five categories of reasoning abilities (arithmetic, logical, commonsense, algorithmic, and planning). In addition to general reasoning benchmarks, several specialized benchmarks have emerged to evaluate some special situations. Overthinking Bench (Cuadron et al., 2025) proposed a framework to assess the extent of overthinking in LLMs. Analyzing 4,018 trajectories revealed that LLMs prefer extended internal reasoning rather than environmental interactions, and it identified several undesirable behavioral patterns, such as Analysis Paralysis, Rogue Actions, and Premature Disengagement. Bag of Tricks (Liu et al., 2025a) evaluates explicitly the impact of TTC techniques on the reasoning abilities of LLMs and presents a benchmark covering six test-time optimization strategies evaluated on eight reasoning tasks. DNA Bench (Hashemi et al., 2025) is a benchmark to assess the over-reasoning problem prevalent in current reasoning models. It comprises 150 adversarial prompts covering four key challenges (e.g., instruction adherence, hallucination avoidance, redundancy filtering, and unanswerable question recognition). DNA Bench highlights that reasoning models often produce redundant or invalid responses to simple yet misleading tasks, causing unnecessary computation and reduced accuracy.

# 5 Discussions and Future Directions

**Efficiency Up Brings Safety Down?** While long CoT has been shown to enhance reasoning capabilities, H-CoT (Kuo et al., 2025) reveals that LRMs can be exploited via extended CoT paths to bypass safety guardrails (Feng et al., 2024a), leading to harmful outputs (Li et al., 2025d). This suggests a tension between safety and efficiency: enhancing safety requires longer, more deliberate reasoning for self-correction, which undermines efficiency, while shorter, efficient reasoning paths may skip critical safety checks. Balancing safety and efficiency remains a crucial challenge for future research in LLM reasoning. Latent reasoning offers a more structured, compact, and controllable process, making it a promising direction for reducing safety risks. Additionally, representation alignment, which constrains internal representations, may serve as a lightweight yet effective strategy for enhancing model safety.

**Efficient Reasoning for Multimodal Large Language Model.** Some efficient reasoning methods can be naturally extended to the multimodal large language model (MLLM) setting. The decomposition strategy discussed in Section 3.3.2, which breaks complex tasks into atomic reasoning units, can also benefit

multimodal reasoning (Xiang et al., 2025a; Hu et al., 2025). Similarly, latent reasoning has shown promise in MLLMs (see Heima in Section 3.1.4). LatentLM (Sun et al., 2024b) further explores this direction by unifying discrete and continuous modalities through latent language modeling. It uses a variational autoencoder (VAE) to encode continuous data into latent vectors and then applies next-token diffusion for autoregressive generation, enabling scalable and efficient multimodal generation. Additionally, efficient reasoning has been extended to typical vision tasks (Wang et al., 2025c; Koksal & Alatan, 2025; Feng et al., 2025; Li et al., 2025c; Ouyang et al., 2023; Shao et al., 2025), offering valuable insights for future research on integrating structured reasoning into vision-centric multimodal applications.

**Break Memory Limitation.** While long reasoning paths bring remarkable performance, they also cause severe memory issues due to long context. PENCIL (Yang et al., 2025a) addresses this by progressively erasing outdated and unimportant reasoning steps during generation. INFTYTHINK (Yan et al., 2025) adopts a segmentation strategy, breaking the reasoning path into shorter fragments and inserting concise intermediate summaries, enabling chunk-wise thinking. OMNIKV (Hao et al., 2025) observes that adjacent layers share highly similar token importance distributions and thus dynamically select key tokens and reuse them across subsequent layers. MCoT (Yang et al., 2024c) models multi-step reasoning as a Markov chain, where each step depends only on the previous one, avoiding the accumulation of long historical states in the KV cache. These methods show the value of memory-efficient designs; future work should pursue lighter architectures (Gu & Dao, 2024; Yuan et al., 2025) and adaptive context management for scalable long-range reasoning.

**Training Efficiency.** Training long reasoning models remains a computationally intensive task. Recent work has aimed to improve training efficiency through both curriculum learning and RL optimization. Curriculum-based approaches, such as Light-R1 (Wen et al., 2025) and FASTCURL (Song et al., 2025), progressively increase task complexity to facilitate stable learning. Light-R1 employs curriculum SFT and multi-stage post-training, achieving strong performance with public datasets. FASTCURL extends this idea by combining curriculum RL with progressive context window extension, enabling efficient training of R1-like models even on limited hardware. On the RL front, DAPO (Yu et al., 2025b) proposes a scalable and open-source RL system, leveraging decoupled clipping and dynamic sampling for improved training stability. AGPO (Li et al., 2025a) addresses critical instability in the popular GRPO (Guo et al., 2025) by introducing a revised advantage estimation that mitigates zero-variance issues. Some coreset methods focus on reducing the quantity of training data. LIMO (Ye et al., 2025) argues that complex reasoning abilities are not learned from scratch but elicited through high-quality samples. By constructing a carefully curated dataset of only 817 reasoning samples, the model trained on this data significantly outperforms those trained on nearly 100K examples. The dataset construction involves filtering out easy problems, retaining challenging ones where advanced models struggle, and performing diversity-based sampling. Similarly, s1 (Muennighoff et al., 2025) constructs a compact dataset of 1,000 examples by jointly optimizing for difficulty, diversity, and quality. Improving training efficiency through algorithmic innovations or data-centric approaches remains a promising direction with substantial room for further exploration.

**Opportunities in Traditional Model Compression.** Traditional model compression techniques offer valuable opportunities for improving reasoning efficiency. Among them, distillation has demonstrated significant potential in enhancing reasoning efficiency. Distillation effectively transfers reasoning abilities from larger models to smaller ones, enabling them to achieve strong reasoning while significantly reducing costs (see Section 3.2.1). Chen et al. (2025b) systematically investigates three key factors that influence the effectiveness of CoT distillation: the granularity of reasoning paths, the format in which reasoning is presented, and the choice of teacher model. These insights offer practical guidance for advancing the distillation of reasoning abilities in small language models. Furthermore, distillation can play a role in other efficient reasoning directions, such as latent reasoning, where it helps compress explicit CoTs into more compact implicit reasoning paths (see Section 3.1.4) and SFT with variable-length CoT data (see Section 3.1.2). Distillation is a promising strategy for efficient reasoning, though there remains room for improvement. Additionally, enhancing the efficiency of the distillation process itself is also a valuable direction for future research. Beyond distillation, other model compression techniques, such as quantization and pruning, also show potential.

Although preliminary pruning experiments were not promising, successful quantization suggests that model compression can maintain reasoning performance while improving efficiency in areas like memory usage.

**Advancing Sustainability through Efficient Reasoning.** As discussed in this work, efficient reasoning techniques contribute to optimizing the efficiency of reasoning models, reducing computational costs, and minimizing resource usage. These approaches help reduce the carbon footprint by lowering the energy requirements and supporting more environmentally friendly practices. As the use of reasoning models grows, adopting more efficient methods can play a crucial role in mitigating the environmental impact. Additionally, these efficiency improvements do not introduce significant negative effects, ensuring the benefits are realized without unintended consequences.

**Comparison with Related Surveys.** Several recent surveys have discussed reasoning models from different angles. For example, Towards Reasoning Era (Chen et al., 2025a) provides a comprehensive overview of long CoT reasoning, focusing primarily on reasoning performance and structure, but does not emphasize efficiency as a central concern. Some surveys (Qu et al., 2025a; Sui et al., 2025b) center on reasoning efficiency. The former (Qu et al., 2025a) organizes methods by stages in the LLM development lifecycle (e.g., pre-training, supervised fine-tuning, reinforcement learning, and inference), offering a broad perspective across the modeling pipeline. The latter (Sui et al., 2025b) classifies approaches based on their core technical mechanisms (e.g., model-based, output-based, and prompt-based), clearly distinguishing the underlying methodological paths. In contrast, our work focuses on how efficiency is achieved during reasoning itself, offering a goal-driven taxonomy centered around making reasoning shorter, smaller, and faster. This structured perspective helps clarify the design space of efficient reasoning and provides clearer guidance for future research.

**Connection between Intrinsic Efficiency Metrics and Hard Performance Metrics.** In practical applications, users are primarily concerned with the efficiency that reasoning methods bring to model deployment and usage, typically measured by hard performance metrics such as time and memory. However, efficient reasoning methods often report token count rather than actual runtime. In practice, token count and latency are strongly correlated. We empirically validated this on Qwen2.5-7B using the MAHT-500 dataset, where we observed a clear positive correlation between token count and latency. The Pearson correlation coefficient was 0.9998 with a near-zero p-value, indicating a statistically significant and nearly perfect linear relationship. Meanwhile, some efficient reasoning methods employ PEFT techniques, such as LoRA, to reduce memory usage and calculation costs during the SFT or RL stages. However, this reduction applies only to the training stage and does not affect memory usage during inference or downstream deployment.

# 6 Conclusion

In conclusion, this survey provides a comprehensive overview of efficient reasoning techniques. We categorize current efforts into three main directions—shorter, smaller, and faster—each addressing reasoning efficiency from a unique perspective: compressing reasoning chains, building small language models with strong reasoning abilities, and accelerating the decoding stage. As reasoning efficiency continues to gain traction, we believe it holds significant promise for enabling scalable and practical deployment of reasoning models across diverse applications, from real-time systems to resource-constrained environments. We hope this survey serves as a valuable foundation for future research and development in this critical and rapidly evolving field.

# Acknowledgments

This project is supported by the Ministry of Education, Singapore, under its Academic Research Fund Tier 2 (Award Number: MOE-T2EP20122-0006).

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

# A Appendix

## A.1 Details for Model Compression

**Quantization.** Quantization improves model efficiency and reduces memory usage by lowering the bit precision of parameters. It is typically categorized into post-training quantization (PTQ) and quantization-aware training (QAT), distinguished by whether retraining is involved. PTQ applies quantization directly to a pre-trained model, while QAT includes a retraining stage to mitigate quantization-induced errors. Quantization can target weights, activations, or both. Advanced methods such as GPTQ (Frantar et al., 2023a), AWQ (Lin et al., 2024), and SmoothQuant (Xiao et al., 2023) further enhance quantization for large language models by reducing activation outliers and minimizing calibration errors.

**Pruning.** Pruning reduces model size and inference latency by eliminating redundant or less important parameters. It can be broadly categorized into unstructured pruning, structured pruning, and semi-structured pruning. Unstructured pruning removes individual weights based on certain criteria, such as magnitude. While it achieves high sparsity, it is often less hardware-friendly due to irregular sparsity patterns. Structured pruning eliminates entire units such as neurons, channels, or attention heads, leading to more regular sparsity patterns that are easier to accelerate in practice. Semi-structured pruning strikes a balance between the two, applying constraints such as N:M sparsity, where only a fixed number of weights are retained in each block. This enables efficient execution on specialized hardware. Recent works (e.g., LLM-Pruner, Dep-Graph) (Ma et al., 2023; Fang et al., 2024; 2023; Feng et al., 2024b), and methods based on importance scores and gradient sensitivity (Wang et al., 2021; Zhang et al., 2021; Tuo & Wang, 2025) have significantly improved the effectiveness and usability of pruning for large models.

**Knowledge Distillation.** Knowledge Distillation (KD) transfers the behavior of a large, well-performing teacher model to a smaller student model by aligning output distributions (e.g., logits or soft labels), intermediate representations, or attention patterns. KD approaches can be categorized as black-box or white-box, depending on whether the student has access only to the teacher's outputs or to internal states as well. Variants like Self-Instruct KD (Wang et al., 2022b) enable the student to inherit reasoning abilities and generalization skills in more flexible settings.

## A.2 Overthinking Example

We provide an example to show the overthinking problem: for the simple question "What is the answer of 1 plus 2?", DeepSeek R1 generates a lengthy response containing 619 tokens, whereas the ideal answer "The

answer is 3." requires only 17 tokens. This highlights the inefficiency brought by overthinking in current models, especially when solving simple problems.

> **Answer (DeepSeek R1)**
>
> Okay, so I need to figure out what 1 plus 2 is. Let me start by recalling basic addition. When you add two numbers together, you're combining their values. So, if I have 1 apple and someone gives me 2 more apples, how many apples do I have in total?
>
> Let me count them out. Starting with 1, then adding another 2. So, 1... then 2, 3. Wait, does that mean 1 plus 2 equals 3? Hmm, that seems right. Let me check another way to make sure I'm not making a mistake. Maybe using my fingers. Hold up one finger on my left hand for the number 1, then two fingers on my right hand for the number 2. Now, if I count all the fingers I'm holding up, that's 1, 2, 3. Yep, that's three fingers. So 1 plus 2 must be 3.
>
> Alternatively, I can think of the number line. Starting at 1, then moving 2 steps to the right. So from 1, moving one step to 2, then another step to 3. That lands me on 3. So that also gives me 3.
>
> Wait, is there another method to verify? Maybe using objects. If I have 1 book and add 2 more books, the total should be 3 books. Let me visualize: one book here, two there. Combine them, count them all: 1, 2, 3. Yep, three books.
>
> I guess all these methods point to the same answer. But just to be thorough, maybe I can use the basic addition facts I learned. The sum of 1 and 2 is one of the fundamental arithmetic facts. From what I remember, $1 + 2$ is indeed 3. There's no carrying over or anything complicated here because both numbers are single-digit.
>
> Another way to look at it is through set theory. If I have a set with 1 element and another set with 2 elements, the union of these sets will have $1 + 2 = 3$ elements, provided there's no overlap. So, assuming the sets are disjoint, the total number of elements is 3.
>
> Also, in terms of equations, if I write it out: $1 + 2 =$ x. Solving for x, I can subtract 1 from both sides, but that might complicate things. Alternatively, just recognizing that addition is commutative, so $1 + 2$ is the same as $2 + 1$, which is 3.
>
> I think I've checked it multiple ways: counting on fingers, using a number line, set theory, and basic arithmetic facts. All confirm that 1 plus 2 equals 3. I don't see any reason to doubt this result. It's straightforward and consistent across different methods. So the answer must be 3.

> **Answer (Expected)**
>
> The answer is 3.

### A.3 Latency Comparison of Efficient Reasoning Methods

Table 5 summarizes representative efficient reasoning methods on GSM8K across different categories, providing a practical overview of efficient reasoning approaches for users.

### A.4 Metric Formulas

#### A.4.1 Carbon Emission

$$\underset{(\text{kgCO}_2\text{eq})}{\text{Carbon Emission}} = \underset{(\text{kWh})}{\text{Energy Consumption}} \times \underset{(\text{gCO}_2\text{eq/kWh})}{\text{Carbon Intensity}} \tag{1}$$

#### A.4.2 Pass@k

$$\text{Pass@}k = 1 - \mathbb{E}_{\text{task}}\left[\frac{\binom{n-c}{k}}{\binom{n}{k}}\right] \tag{2}$$

where $n$ is the number of sampled outputs and $c$ is the number of correct ones.

Table 5: Overview of efficient reasoning methods on GSM8K. The speedup ratio is computed mainly through latency comparison, except for Self-Calibration, where sampling count (S.) is used as a proxy.

| Category / Type | Methods | Training Scheme | Accuracy | Base Model | Speedup |
|---|---|---|---|---|---|
| Shorter / Routing | Self-REF | SFT (LoRA) | **81.60%** | LLaMA3-8B-I | 1.3 × |
| Smaller / KD | SKIntern | Distillation (LoRA) | 62.50% | LLaMA3-8B-I | - |
| Faster / Efficient self-consistency | Path-Consistency | Training-free | 67.80% | LLaMA3-8B-I | 1.2 × |
| Shorter / SFT | CoT-Valve | Progressive SFT (LoRA) | 87.30% | LLaMA3.1-8B-I | 1.7 × |
| Shorter / SFT | TokenSkip | SFT (LoRA) | 78.20% | LLaMA3.1-8B-I | 1.7 - 1.8 × |
| Shorter / SFT | TALE-PT | SFT (LoRA) | 78.57% | LLaMA3.1-8B-I | 1.7 × |
| Shorter / Latent reasoning | SoftCoT | SFT (Freeze FT) | 81.03% | LLaMA3.1-8B-I | 4.0 - 5.0 × |
| Shorter / Latent reasoning | LightThinker | SFT (Full FT) | 88.25% | LLaMA3.1-8B-I | up to 1.4 × |
| Shorter / Latent reasoning | Token Assorted | SFT (Full FT) | 84.10% | LLaMA3.1-8B-I | 1.2 × |
| Smaller / KD | Mix | Mixed distillation (Full FT & LoRA) | 81.40% | LLaMA3.1-8B-I | - |
| Smaller / KD | DLCoT | Distillation (Full FT) | **93.60%** | LLaMA3.1-8B-I | - |
| Faster / Efficient sampling | $\phi$-Decoding | Training-free | 86.58% | LLaMA3.1-8B-I | 2.8 × |
| Faster / Efficient self-consistency | Self-Calibration | SFT (Full FT) | 80.43% | LLaMA3.1-8B-I | 16.7 × (S.) |

### A.4.3 Pass∧k

$$Pass \wedge k = \mathbb{E}_{\text{task}} \left[ \frac{\binom{c}{k}}{\binom{n}{k}} \right] \tag{3}$$

where $n$ is the number of sampled outputs and $c$ is the number of correct ones.

### A.4.4 G-Pass@k

$$\text{G-Pass@}k_\tau = \mathbb{E}_{\text{task}} \left[ \sum_{j=\lceil \tau k \rceil}^{c} \frac{\binom{c}{j}\binom{n-c}{k-j}}{\binom{n}{k}} \right] \tag{4}$$

where $n$ is the number of sampled outputs, $c$ is the number of correct ones, and $\tau$ is a tolerance threshold that represents the minimum proportion of correct responses among the $k$ outputs.

$$\text{mG-Pass@}k_\tau = \frac{2}{k} \sum_{i=\lceil 0.5k \rceil+1}^{k} \text{G-Pass@}k_{\frac{i}{k}} \tag{5}$$

### A.4.5 Outcome and Process Efficiency Metric

**Outcome Efficiency Metric:**

$$\xi_O = \frac{1}{N} \sum_{i=1}^{N} \sigma_i \frac{\hat{T}_i}{T_i} \tag{6}$$

where $N$ is the number of instances, $T_i$ denotes the total number of tokens generated for instance $i$, $\hat{T}_i$ is the number of tokens until the first correct answer, and $\sigma_i$ indicates correctness:

$$\sigma_i = \begin{cases} 1, & \text{if at least one solution is correct} \\ 0, & \text{otherwise} \end{cases}$$

**Process Efficiency Metric:**

$$\xi_P = \frac{1}{N} \sum_{i=1}^{N} \frac{D_i}{T_i} \tag{7}$$

where $D_i$ represents tokens contributing to solution diversity, defined as:

$$D_i = \sum_{m=1}^{M} \tau_i^m T_i^m$$

where $T_i^m$ is the token count of the $m$-th solution for instance $i$, and $\tau_i^m$ denotes whether the solution introduces a new reasoning strategy:

$$\tau_i^m = \begin{cases} 1, & \text{if solution } m \text{ is distinct in reasoning} \\ 0, & \text{otherwise} \end{cases}$$

### A.4.6 Reasoning Boundary (RB)

$$B_{Acc=K_1}(t|m) = \sup_d \{d \mid \text{Acc}(t|d,m) = K_1\} \tag{8}$$

where $t$ denotes a specific reasoning task, $m$ represents the evaluated language model, $d$ indicates the difficulty level of the task, $\text{Acc}(t|d,m)$ is the accuracy of model $m$ on task $t$ with difficulty $d$, $K_1$ is a predefined accuracy threshold, sup denotes the supremum (least upper bound) over the set of difficulty levels satisfying the accuracy condition.

### A.4.7 Underthinking Metric

$$\xi_{\text{UT}} = \frac{1}{N} \sum_{i=1}^{N} \left(1 - \frac{\hat{T}_i}{T_i}\right) \tag{9}$$

where $N$ is the number of incorrect response instances in the test set, $T_i$ is the total number of tokens in the $i$-th incorrect response, $\hat{T}_i$ is the number of tokens from the beginning of the $i$-th response up to and including the first correct thought.

### A.4.8 Accuracy Efficiency Score

$$\Delta\text{Length} = \frac{\text{Length}_{\text{baseline}} - \text{Length}_{\text{model}}}{\text{Length}_{\text{baseline}}},$$
$$\Delta\text{Acc} = \frac{\text{Acc}_{\text{model}} - \text{Acc}_{\text{baseline}}}{\text{Acc}_{\text{baseline}}}$$

Then, the AES is computed as:

$$\text{AES} = \begin{cases} \alpha \cdot \Delta\text{Length} + \beta \cdot |\Delta\text{Acc}|, & \text{if } \Delta\text{Acc} \geq 0 \\ \alpha \cdot \Delta\text{Length} - \gamma \cdot |\Delta\text{Acc}|, & \text{if } \Delta\text{Acc} < 0 \end{cases}$$

where $\alpha > 0$, $\beta > 0$, and $\gamma > 0$ are weighting factors. The default values $\alpha = 1$, $\beta = 3$, and $\gamma = 5$ are used to emphasize penalizing accuracy drop more heavily than rewarding accuracy improvement.

### A.5 Complete List of Datasets and Benchmarks

A complete list of the datasets and benchmarks used in this area is summarized in Table 6, offering researchers an organized reference for efficient reasoning evaluation.

Table 6: Full List of Datasets and Benchmarks.

| Type | Name | Task / Target | Source |
|---|---|---|---|
| *Datasets* | GSM8K | Math | HuggingFace Dataset |
| | MATH & MATH-500 | Math | HuggingFace Dataset |
| | AIME | Math | HuggingFace Dataset |
| | AMC | Math | HuggingFace Dataset |
| | AQuA | Math | HuggingFace Dataset |
| | ProntoQA | Logical | GitHub |
| | StrategyQA | Common sense | HuggingFace Dataset |
| | HotPotQA | Common sense | HuggingFace Dataset |
| | Game of 24 | Algorithmic | GitHub |
| | Bin Packing | Algorithmic | GitHub |
| | BlocksWorld | Planning | HuggingFace Dataset |
| | Rubik's Cube | Planning | GitHub |
| | Trip Plan | Planning | GitHub |
| | Calendar Plan | Planning | GitHub |
| *Benchmarks* | Sys2Bench | General reasoning | GitHub |
| | Overthinking Bench | Overthinking | GitHub |
| | Bag of Tricks | Test-time computation (TTC) | GitHub |
| | DNA Bench | Over-reasoning | - |

