# OpenReview forum: "Efficient Reasoning Models: A Survey"
_TMLR — Accepted by TMLR_

### Review · Reviewer_hR1o · 2025-06-02

**Summary Of Contributions:**

This survey categorises LLM/LRM research on efficiency in three main categories: one for methods that aim at shortening the CoT; one for methods that aim at compressing the models; and one for the methods that try to make the decoding faster.

**Audience:**

Yes

**Broader Impact Concerns:**

No.

**Claims And Evidence:**

Yes

**Requested Changes:**

My two main points of concern should be tackled.

**Strengths And Weaknesses:**

Overall I like the three main categories, and I think the authors did a nice job in reviewing the state of the art, especially considering the huge number of works in this topic. However I feel that sometimes the “reasoning” aspect is almost secondary. Especially in the third category, it sounds like it is about efficiency in general, not really focused on reasoning. Of course, every bit helps, but then the premises of the paper are a bit misleading the reader.

I have two main problems with this survey:

1. The background section is basically non-existent. Concepts that are crucial for the paper are not properly introduced (e.g. pruning, quantization, TTS, RL in the context of LLMs, etc..), and this makes the work not very self-contained. From a survey I would expect it to give more of the necessary background to understand the at least the important distinctions among the different methods. I suspect is not easy to find the right level of details to give to the reader, but that is part of the effort in writing a survey.
2. The text is redundant and it often feels like a “bibliography dump”. There are long lists of works with a short sentence explaining what they do, but not many insights or intuitions that can be extracted / generalised. For example the tables are just quickly referenced in the text, and the only detailed explanation is the caption that basically just describes the columns. Then, all the effort of extracting useful insights / patterns from them is left to the reader. If the space is a problem I would recommend removing some of the “redundancy”. For example a lot of space is devoted to explaining what will be discussed in the next sections, and then for each section there some text describing in more details what was already anticipated before, and then each point repeats again what was anticipated. I understand the need of repeating some concepts to make them clear in the reader’s mind, but this feels a bit too much. For example, the beginning of “Prompt-guided Efficient Reasoning.” in 3.1.3 is almost the same text explained at the beginning of 3.1.3 just a few lines above. That is in turn repeated from the introduction at the beginning of 3.1, that is in turn quite repeated from the general section 3 introduction.

Despite these points, I find the paper well written with nice figures to visualise the proposed categorisation. Especially Figure 2 is nice and useful, also because it is consistent with the structure of the paper.

---

> ### Author Response · Authors · 2025-07-19
> **Response**
>
> > Affirmation of the categorization, inclusion of a large number of works, and our figures.
>
> We sincerely thank the reviewer for the positive feedback on our categorization and the overall review of the state of the art. We appreciate the reviewer’s thoughtful reading and valuable comments, which are very helpful in improving the quality of our paper. We will revise and refine the manuscript accordingly. We mark the revised contents in **red** and deleted contents with a **line**.
>
> > The core of reasoning is not highlighted in some parts of the article.
>
> We thank the reviewer for the helpful comment. This survey covers two main types of techniques: reasoning-specific methods (e.g., CoT compression) and general acceleration techniques (e.g., model compression and inference speed-up). While some are broadly applicable, we focus on their role in improving reasoning efficiency. In the third category (faster), we distinguish two main lines of work. The first focuses on more efficient use of TTS strategies that enhance reasoning performance, primarily by reducing the number of samples or by early termination of bad reasoning chains (Section 3.3.1). The second centers on decoding acceleration techniques designed for reasoning models, such as reducing redundant computation through decompositional reasoning, and dynamically allocating compute based on problem complexity (Section 3.3.2). In response, we clarified in the abstract that the third category focuses on reasoning efficiency. We also revised the descriptions of methods like SoT [1], SGD [2], and RSD [3] in Section 3.3 to better highlight their relevance to reasoning. Please see the revised manuscript for details.
>
> > Problem 1: The background section is insufficient and lacks some essential explanations of key concepts.
>
> We sincerely thank the reviewer for the constructive comment. We fully agree that a more complete background section is essential for making the survey self-contained and accessible. In response, we have significantly expanded the background section to introduce key concepts relevant to efficient reasoning, including underlying techniques of reasoning models (Section 2.2), TTS (Section 2.3), and model compression (Section 2.4, Appendix A.1). Additionally, we added intuitive token count statistics from DeepSeek R1 series in the ``Why We Need Efficient Reasoning'' section (Section 2.5) to help readers better understand the necessity of pursuing reasoning efficiency.
>
> > Problem 2: The text is redundant with limited insights, often resembling a list of works rather than providing meaningful analysis or synthesis.
>
> We sincerely thank the reviewer for the insightful comments. To address the concern on textual redundancy, we carefully reviewed the entire manuscript and removed repetitive expressions, with particular focus on Section 3. Please refer to the revised version for detailed edits. Regarding the limited insight extraction, we conducted a more in-depth analysis of the tables and incorporated key observations in the paper. For example, Tables 2 and 3 show that SFT-based methods commonly use PEFT techniques like LoRA, while RL-based methods often rely on full fine-tuning. This suggests that RL-based approaches may require more extensive updates to the parameter space, making lightweight adaptation methods like LoRA less effective. Moreover, in terms of actual speedup, latent reasoning exhibits a higher upper bound on token compression, suggesting that implicit representations enable more effective compression than explicit reasoning chains (Section 3.1 & 3.2 & 3.3). We further expanded Section 5 to include additional insights and potential future directions. For instance, we discussed the advantages of latent reasoning in enhancing model safety and the potential of representation alignment as a lightweight safety-enhancement strategy. We also explored the extension of efficient reasoning to multimodal settings, referencing recent work such as PixelThink [4], which demonstrates the application of reasoning models to classical vision tasks (e.g., segmentation) while maintaining efficiency. Additionally, we discussed the development potential of memory-efficient architectures for reasoning models and outlined future directions for improving training efficiency.
>
> References:
>
> [1] Skeleton-of-thought: Prompting LLMs for Efficient Parallel Generation
>
> [2] Adaptive Skeleton Graph Decoding
>
> [3] Reward-Guided Speculative Decoding for Efficient LLM Reasoning
>
> [4] PixelThink: Towards Efficient Chain-of-Pixel Reasoning

---

### Review · Reviewer_whuz · 2025-06-09

**Summary Of Contributions:**

This survey provides a comprehensive overview of recent advancements in making large reasoning models more efficient. The paper addresses the "urgent need for effective acceleration" stemming from the computational overhead introduced by the "slow-thinking" paradigm of models generating long cots

**Audience:**

Yes

**Claims And Evidence:**

Yes

**Requested Changes:**

Please see above weaknesses and questions. Overall, more appropriate discussion is required

**Strengths And Weaknesses:**

Strengths and weaknesses

Strengths
- The paper's organization of a complex and rapidly evolving research area into the intuitive "Shorter, Smaller, Faster" framework. This categorization is logical, easy to grasp, and effectively delineates the primary approaches to efficiency, making the landscape accessible to both newcomers and experts.
- The authors have compiled an extensive and up-to-date collection of research, with a significant number of references from 2024 and 2025. The survey thoroughly covers a wide spectrum of techniques, from model compression and reinforcement learning to prompt engineering and novel decoding strategies, ensuring a holistic perspective.


Weaknesses
- My major concern is that, this is a rapidly evolving area with many preprint papers emerging every day. Under this context, I wonder it is rather difficult to make solid conclusions considering these manuscripts. The papers' claims could be opposite and hard to tell them apart. In short, it may not be a good time to write such a survey while most research is still preprints and not peer-reviewed.
- While the taxonomy is a major strength, the paper could benefit from a more dedicated discussion on the intersections between the three categories. Many methods are multi-faceted; for example, knowledge distillation (smaller) could be used to achieve latent reasoning (shorter). A deeper synthesis exploring how these strategies can be combined for even greater gains would enhance the paper's practical impact.
- More background inclusion. This paper could benefit from a more detailed background section introducing the current status of large reasoning models, such as o1 and R1. This could help readers understand how long cots are elicited and why and where they are useful for advanced reasoning.
- Another line of work, model architecture for efficient reasoning should also be considered. Works such as NSA[1], MOBA[2], Mamba[3] allow more efficient inference thus improved reasoning efficiency as well.
- Some related surveys[4,5] should be cited and discussed, especially where does this survey differs from them and the unique contribution it made.


Reference

[1] Native Sparse Attention: Hardware-Aligned and Natively Trainable Sparse Attention

[2] MoBA: Mixture of Block Attention for Long-Context LLMs

[3] Mamba: Linear-Time Sequence Modeling with Selective State Spaces

[4] A survey of efficient reasoning for large reasoning models: Language, multimodality, and beyond

[5] Towards reasoning era: A survey of long chain-of-thought for reasoning large language models.

---

> ### Author Response · Authors · 2025-07-19
> **Response**
>
> > The categorization is well-founded, and the covered methods and literature are comprehensive.
>
> We sincerely appreciate the reviewer’s recognition of our taxonomy and thoughtful evaluation of our work. We are also grateful for the valuable comments, which have helped us refine and improve the paper. We will revise the manuscript accordingly to address the comments. We mark the revised contents in **red** and deleted contents with a **line**.
>
> > Weakness 1: Concern about the maturity of the field and the reliability of preprint-dominated literature.
>
> We sincerely appreciate the reviewer for the valuable comments. Efficient reasoning is in a rapidly developing stage. We believe that a timely survey can play a valuable role by summarizing the current landscape and offering potential insights for future research directions. Given the influx of new work, including many preprints, such a summary may help both newcomers and active researchers quickly grasp emerging trends and challenges, facilitating more informed and impactful progress.
>
> > Weakness 2: Add discussion on the intersections between the three categories.
>
> We sincerely thank the reviewer for the insightful comment. We fully agree that discussing the intersections between different strategies can enhance the paper's depth. In Section 3.4, we have added a discussion on Intersections and Synergies Across Efficient Strategies, highlighting the broad applicability of distillation, including its role in latent reasoning and SFT-based methods. We also describe how combining SFT and RL enables adaptive reasoning, where models dynamically switch between fast and slow thinking modes based on problem difficulty, as demonstrated by [1,2].
>
> > Weakness 3: More background inclusion.
>
> We sincerely thank the reviewer for the constructive comment. A more detailed background section helps readers quickly grasp the core topic and better understand the subsequent content. In response, we have expanded Section 2.2 with an overview of large reasoning models and their underlying techniques. We also enhanced Section 2.5 with empirical data on CoT lengths to illustrate the inefficiency of the “slow-thinking” paradigm. Furthermore, we introduced key concepts relevant to our survey, such as TTS (Section 2.3) and model compression (Section 2.4, Appendix A.1), to improve the self-contained nature of the background.
>
> > Weakness 4: Add another line of work - model architecture for efficient reasoning.
>
> We sincerely thank the reviewer for the valuable comment. We agree that model architecture plays a key role in reasoning efficiency. We categorize these efficient model design approaches under model-focused distillation, as their core mechanism involves transferring reasoning capabilities via distillation. In addition to NSA, MoBA, and Mamba, we have also included reasoning-oriented methods such as M1 [3], which builds reasoning models using lightweight architectures like Mamba.
>
> > Weakness 5: Some related surveys should be cited and discussed.
>
> We sincerely thank the reviewer for the helpful comment and agree that discussing related surveys improves the paper. In Section 5, we have added a comparative discussion, including both the survey suggested by the reviewer and the Stop Overthinking survey [4].
>
> References:
>
> [1] Thinkless: LLM Learns When to Think
>
> [2] ARM: Adaptive Reasoning Model
>
> [3] M1: Towards Scalable Test-Time Compute with Mamba Reasoning Models
>
> [4] Stop Overthinking: A Survey on Efficient Reasoning for Large Language Models

---

### Review · Reviewer_ur1E · 2025-07-24

**Summary Of Contributions:**

This is a survey paper on methods for making reasoning language models more efficient to use. The authors identify three key axes of efficiency: parameter-efficiency ("smaller"), token-efficiency ("shorter") and decoding efficiency ("faster"), survey a long list of works and main ideas in improving each of these dimensions, compile a list of key metrics, datasets and benchmarks relevant to the field, and end with a discussion of open problems (including safety, multimodality, memory requirements of long reasoning chains, and training efficiency).

**Audience:**

Yes

**Claims And Evidence:**

Yes

**Requested Changes:**

See two weaknesses above, summarized here:

1- For a paper focused on efficiency, I believe it is quite important to connect the intrinsic efficiency metrics (e.g., tokens, or fine-tuning strategy) to extrinsic, hard performance metrics (like time and memory).
2- It would also be helpful to have some sense of comparison across the categories, using these same hard metrics. Perhaps looking at a few works that share the same base model, and showing how the efficiency improvements compare on that same model, could be a way to go about this.

I'm inclined to request these two to recommend the paper for acceptance, but I'm open to discussing with the authors whether they think they are reasonable. If not, I think the paper would need at least a discussion of why these comparisons are hard to make.

**Strengths And Weaknesses:**

The paper is both comprehensive and focused (on efficiency). It includes a significant breadth of the works that are mentioned in each category. The taxonomy that is proposed is ample and is able to cleanly capture a quite broad body of existing work in this new field (e.g., in Figure 2). The field of reasoning models generally is rapidly evolving and gaining significant attention, so an attempt to survey a key dimension in the field (here, efficiency) is timely and I believe a useful contribution for the field.

However, I did find the paper lacking in helping me gain a view of the field that is richer than the list of papers. While the three dimensions that are proposed are interesting, I think there are two weak aspects in how they're used in the paper:

1- The axes of "token efficiency", "model size" and "decoding efficiency" are not intrinsic metrics that users of reasoning models care about. They are proxies for the actually concrete efficiency metrics -- memory and time. Essentially, how much memory do I need to run the model (or train/fine-tune, but that's listed in the future directions discussion only), and how fast do I get complete responses to the problems on a given hardware. Ultimately, users care about these metrics, and improving tokens, model size, and decoding algorithms are only a way to attempt to impact these concrete metrics. But I did not find the paper particularly elucidating with respect to these metrics. I believe it would be very valuable to readers to know how the proposed axes impact these concrete efficiency metrics -- I believe many of the cited works might provide data points on this. This will also make it easier to compare the impact across categories (for instance, how much faster can you get good responses by prompting the model to "be concise", vs by changing the decoding strategy? Time and memory are the concrete criteria I'd use to select which one to use, assuming accuracy remains similar).

2- Related to this (and a suggestion): I found it difficult to get a sense of which of the ideas in existing work have the largest potential to impact efficiency. I wouldn't be surprised if some are quite drastic (say, 2-3x reduction in run time) and some are marginal (say, 5%). Section 2.5 now makes an attempt to make this concrete both by showing numbers obtained with DeepSeek-R1, both on a toy example (2 + 2) and on AIME 24. These examples weren't followed up on in the rest of the paper, but I think this can be one way to make the survey much richer. Say you take any base model (doesn't have to be R1, which is quite large) and then attempt some representative method from several of the cited categories (e.g. prompting, decoding, reasoning summarization, etc) and show how that idea impacts efficiency. Even if this is just in a small number of examples, this would help readers get a concrete sense of what is the actual potential of existing works for practical use cases.

---

> ### Author Response · Authors · 2025-08-07
> **Response to W1**
>
> > The taxonomy is solid and well-justified, with broad and thorough coverage of relevant methods and literature. Such a timely survey is valuable for a rapidly evolving field.
>
> We sincerely thank the reviewer for recognizing the strength of our taxonomy and for the thoughtful feedback. The comments are greatly appreciated and have been instrumental in helping us further refine the paper. We will revise the manuscript to reflect these suggestions. We mark the revised contents in **red** and deleted contents with a **line**.
>
> ---
>
> > **W1:** For a paper focused on efficiency, I believe it is quite important to connect the intrinsic efficiency metrics (e.g., tokens, or fine-tuning strategy) to extrinsic, hard performance metrics (like time and memory).
>
> We fully acknowledge and appreciate the reviewer’s emphasis on the importance of hard metrics such as **memory and time**. To emphasize time, we have summarized the latency speedups of representative methods within the same category in Tables `2` and `4`. Inspired by the reviewer‘s suggestion in `W2`, we further provide a summary of popular methods across categories, **comparing latency speedup** (see the response to `W2` for detailed quantitative results). Meanwhile, we highlight the importance of **memory efficiency** and **provide a focused analysis** in Sec. `3.2.2`. Details as follows.
>
> **<Memory>** For memory optimization, several papers in the 'smaller' category explore the effects of model compression methods such as GPTQ [`1`], AWQ [`2`], and SparseGPT [`3`] on reasoning models. From the provided results, we observe that quantization can reduce GPU memory usage by over 75% while maintaining original performance on certain large models, and quantized versions of large models often outperform standalone small models, providing benefits in both memory efficiency and overall performance.
>
> **<Time (latency)>** In this work, we indeed focus on **time** as the key hard metric, specifically targeting inference efficiency like shorter output and faster decoding. As a quantitative metric, we use **latency** as the evaluation criterion for time, as detailed in the first paragraph. Besides, most existing methods adopt token count to evaluate efficiency, as it is hardware-agnostic and thus ensures fair comparison across models, while latency is heavily influenced by hardware configurations, making it less suitable for consistent benchmarking across different methods.
>
> We agree that the reviewer highlights the connection between *'intrinsic efficiency metrics'* and *'hard performance metrics'*. To this end, we first conduct experiments using `Qwen2.5-7B` on the MATH-500 dataset, recording both token count and corresponding latency to calculate their correlation, and observe a strong positive linear correlation between them (10 sampling points are presented below). In addition, we note that several involved methods **adopt PEFT techniques** to significantly reduce memory usage and computational costs during the SFT and RL stages. While effective for improving training efficiency, these techniques have minimal impact on the memory footprint during inference, and thus do not substantially affect the efficiency experienced by end users.
>
> | **Sample ID** | **Token Count** | **Latency (s)** |
> |-|-|-|
> | 0 | 79 | 0.6725 |
> | 1 | 103 | 0.8460 |
> | 2 | 182 | 1.4812 |
> | 3 | 202 | 1.6459 |
> | 4 | 247 | 2.0169 |
> | 5 | 431 | 3.4769 |
> | 6 | 579 | 4.6479 |
> | 7 | 613 | 4.9372 |
> | 8 | 2336 | 19.2317 |
> | 9 | 8174 | 67.9248 |
>
> We have updated the manuscript to include the above discussions accordingly (refer to Sec. `3(.2.2)`, Table `5`, and Sec. `5` of our revised manuscript). We sincerely thank the reviewer again for highlighting the importance of hard metrics and helping us further improve the quality of our work.
>
> ---
>
> **References:**
>
> - [`1`] GPTQ: Accurate Post-Training Quantization for Generative Pre-trained Transformers
> - [`2`] AWQ: Activation-aware Weight Quantization for LLM Compression and Acceleration
> - [`3`] SparseGPT: Massive Language Models Can Be Accurately Pruned in One-Shot

---

> ### Author Response · Authors · 2025-08-07
> **Response to W2**
>
> > **W2:** It would also be helpful to have some sense of comparison across the categories, using these same hard metrics. Perhaps looking at a few works that share the same base model, and showing how the efficiency improvements compare on that same model, could be a way to go about this.
>
> We sincerely thank the reviewer for this valuable suggestion. Inspired by the reviewer’s suggestion, we compared **representative methods** from **different categories** under the **same base model** to provide practical guidance for real-world use. Using GSM8K as the evaluation dataset due to its broad coverage, we report their **actual latency speedups**.
>
> | **Category / Type**  | **Methods** | **Training Scheme** | **Accuracy**  | **Base Model** | **Speedup** |
> |-|-|-|-|-|-|
> | Shorter / Routing | Self-REF | SFT (LoRA) | **81.60%** | LLaMA3-8B-I | 1.3$\times$ |
> | Smaller / KD | SKIntern | Distillation (LoRA) | 62.50% | LLaMA3-8B-I | - |
> | Faster / Efficient self-consistency | Path-Consistency | Training-free | 67.80% | LLaMA3-8B-I | 1.2$\times$ |
> | Shorter / SFT | CoT-Valve | Progressive SFT (LoRA) | 87.30% | LLaMA3.1-8B-I | 1.7$\times$ |
> | Shorter / SFT | TokenSkip | SFT (LoRA) | 78.20% | LLaMA3.1-8B-I | 1.7 - 1.8$\times$ |
> | Shorter / SFT | TALE-PT | SFT (LoRA) | 78.57% | LLaMA3.1-8B-I | 1.7$\times$ |
> | Shorter / Latent reasoning | SoftCoT | SFT (Freeze FT) | 81.03% | LLaMA3.1-8B-I | 4.0 - 5.0$\times$ |
> | Shorter / Latent reasoning | LightThinker | SFT (Full FT) | 88.25% | LLaMA3.1-8B-I | up to 1.4$\times$ |
> | Shorter / Latent reasoning | Token Assorted | SFT (Full FT) | 84.10% | LLaMA3.1-8B-I | 1.2$\times$ |
> | Smaller / KD | Mix | Mixed distill (Full FT & LoRA)| 81.40% | LLaMA3.1-8B-I | - |
> | Smaller / KD | DLCoT | Distillation (Full FT) | **93.60%** | LLaMA3.1-8B-I | - |
> | Faster / Efficient sampling | $\phi$-Decoding | Training-free | 86.58% | LLaMA3.1-8B-I | 2.8$\times$ |
> | Faster / Efficient self-consistency| Self-Calibration | SFT (Full FT) | 80.43% | LLaMA3.1-8B-I    | 16.7$\times$ (*S.*) |
>
> > **Note:** The speedup ratio is computed mainly through latency comparison, except for *Self-Calibration*, where sampling count (*S.*) is used as a proxy.
>
> We observe that DLCoT, which distills a large amount of long CoT data, achieves the highest accuracy among these methods. When considering both accuracy and speedup, methods in the *shorter* category demonstrate a more favorable trade-off. In particular, latent reasoning approaches exhibit the potential for achieving a higher degree of speedup.
>
> We have incorporated the above summary into the revised manuscript (refer to Table `5` in Sec. `A.3` of our revised manuscript). We sincerely thank the reviewer once again for the insightful suggestion, which helps enrich and strengthen our work.

---

### Decision · Action_Editor_t3Nf · 2025-09-13

**Recommendation:** Accept as is

**Audience:**

Yes

**Audience Explanation:**

This submission is a thorough survey for efficient reasoning models. It can be a valuable resource for the readers who are interested in this area of research.

**Claims And Evidence:**

Yes

**Claims Explanation:**

All reviewers show their satisfaction towards the authors’ revision.